# A Robust Minimization-Based Framework for Cyclogeostrophic Ocean Surface Current Retrieval

Vadim Bertrand[1], Julien Le Sommer[1], Victor Vianna Zaia De Almeida[1], Adeline Samson[2], and Emmanuel Cosme[1]

[1]Univ. Grenoble Alpes, CNRS, IRD, Grenoble INP, IGE, Grenoble, F-38000, France
[2]Univ. Grenoble Alpes, CNRS, Grenoble INP, LJK, Grenoble, F-38000, France

**Correspondence:** Vadim Bertrand (vadim.bertrand@univ-grenoble-alpes.fr)

**Abstract.** Estimations of surface currents at submesoscales (1–50 km) are crucial for operational applications and environmental monitoring, yet accurately deriving them from satellite observations remains a challenge. While the geostrophic approximation has long been used to infer ocean surface currents from Sea Surface Height (SSH), it neglects nonlinear advection, which can become significant at submesoscales. To address this limitation, we present a robust and efficient minimization-based method for inverting the cyclogeostrophic balance equation, implemented in the open-source Python library `jaxparrow`. Unlike the traditional fixed-point approach, our method reformulates the inversion as a minimization problem, providing stable estimates even in regions where a cyclogeostrophic solution may not exist. Using a submesoscale-permitting model simulation and both DUACS and the high-resolution NeurOST SSH products, we demonstrate that cyclogeostrophic corrections become increasingly relevant at finer spatial scales. Validation against drifter-derived velocities shows that our approach consistently improves current estimates in energetic regions, reducing errors by up to 20 % compared to geostrophy alone in energetic regions of the global ocean. These results support the systematic inclusion of cyclogeostrophic inversion in the analysis of high-resolution SSH fields.

## 1  Introduction

Surface ocean currents play a critical role in a wide range of environmental and operational processes (Röhrs et al., 2023). At spatial scales from 1 to 50 km—commonly referred to as submesoscales—these currents influence the exchange of energy between the ocean and atmosphere, with important implications for climate studies (Hewitt et al., 2022). They are also essential for numerous practical applications, including offshore operations, renewable energy development (Ferreira et al., 2016), and the forecasting of object trajectories in the ocean. Accurate surface current information supports search-and-rescue missions, iceberg tracking, and the management of marine debris and oil spills (Breivik et al., 2013; Keghouche et al., 2009; Trinanes et al., 2016; De Dominicis et al., 2016). Additionally, submesoscale dynamics contribute to vertical mixing in the upper ocean, affecting biological productivity and the transport of nutrients and plankton, which are key components of marine ecosystems (Mahadevan, 2016; Lévy et al., 2018).

Satellite observations of Sea Surface Height (SSH) and Sea Surface Temperature (SST) both provide valuable insights into surface currents and fine-scale ocean dynamics. Since the 90's, satellite altimetry has provided SSH observations that are then processed into global gridded maps (Le Traon and Dibarboure, 1999) from which geostrophic velocities can be derived (Le Traon and Dibarboure, 2002). The effective resolution of these maps are estimated at nearly 200 km at mid-latitudes (Taburet et al., 2019), keeping the submesoscale spectrum invisible to us. The Surface Water and Ocean Topography mission (SWOT, Fu, 2008; Fu et al., 2012), launched in 2022, has been designed to increase the spatial resolution of earlier altimeters and reach 15 km of effective resolution in the satellite swath (Morrow et al., 2019; Nencioli et al., 2025; Wang et al., 2025). Complementary to altimetry, SST provides high-resolution snapshots of ocean surface structures, revealing submesoscale features which are not observed by conventional altimeters. Many research efforts are presently under way to derive global maps of SSH and currents with a resolution that would enable the observation of the high-wavenumber portion of the spectrum of the mesoscale dynamics by synthesizing classical altimetry with SWOT (Le Guillou et al., 2021; Ubelmann et al., 2021; Ballarotta et al., 2023; Le Guillou et al., 2023; Xiao et al., 2023) and/or SST (Buongiorno Nardelli et al., 2022; Archambault et al., 2023; Fablet et al., 2024; Le Guillou et al., 2025; Martin et al., 2024). In addition, there is a growing interest within the SWOT community in moving beyond the geostrophic approximation when exploiting the high-resolution 2D SSH fields of the SWOT swath (Archer et al., 2025; Wang et al., 2025; Zhang and Callies, 2025; Tranchant et al., 2025; Tchonang et al., 2025; Dù et al., 2025).

Under some dynamical conditions, accurately deriving ocean surface currents from high-resolution SSH images or maps requires using the cyclogeostrophic balance approximation rather than the usual geostrophic approximation. To introduce these relationships, we start from the horizontal momentum equation in a rotating frame:

$$\frac{\partial \boldsymbol{u}}{\partial t} + (\boldsymbol{u} \cdot \nabla)\boldsymbol{u} + f\boldsymbol{k} \wedge \boldsymbol{u} = -g\nabla\eta + \boldsymbol{R} \tag{1}$$

where $\boldsymbol{u}$ is the horizontal velocity, $f$ the Coriolis parameter, $g$ the gravity, $\eta$ the SSH, $\boldsymbol{k}$ the vertical unit vector, and $\boldsymbol{R}$ collects frictional and unresolved processes (e.g. horizontal and vertical mixing, wind stress-driven Ekman current, and other ageostrophic contributions). Bold fonts indicate vectors. The geostrophic balance results from neglecting the local acceleration, the nonlinear advective term, and the residual term:

$$f\boldsymbol{k} \wedge \boldsymbol{u}_g = -g\nabla\eta \tag{2}$$

where $\boldsymbol{u}_g$ is the geostrophic velocity. By retaining the nonlinear advective term while still neglecting the local acceleration and residual processes, one obtains the cyclogeostrophic balance:

$$(\boldsymbol{u}_{cg} \cdot \nabla)\boldsymbol{u}_{cg} + f\boldsymbol{k} \wedge \boldsymbol{u}_{cg} = -g\nabla\eta \tag{3}$$

where $\boldsymbol{u}_{cg}$ is the cyclogeostrophic velocity. This equation extends the usual geostrophic balance equation when the Rossby number $Ro$, defined as the ratio between the scales of the advective term and the Coriolis term, approaches 1. This "$Ro \approx 1$" regime actually characterizes the submesoscale regime (McWilliams, 2019; Taylor and Thompson, 2023). Cyclogeostrophic currents can substantially differ from geostrophic currents in some regions such as the Mozambique channel (Penven et al.,

2014), the Mediterranean sea (Ioannou et al., 2019), and the Antarctic Circumpolar Current (Tranchant et al., 2025). A global assessment by Cao et al. (2023) further indicates that important differences are also expected in the Gulf Stream, the Agulhas Current, and the Kuroshio Current.

Several methods to solve the cyclogeostrophic inverse problem have been proposed in the past literature but they all exhibit drawbacks, and publicly available, well maintained implementations are missing. Penven et al. (2014) provides a review of these methods. The most widely employed, proposed by Arnason et al. (1962) and Endlich (1961), solves the cyclogeostrophic balance by iteratively updating the velocity through a fixed-point relation that adds the nonlinear advective correction to the geostrophic velocity (see Eq. 7). Unfortunately, Arnason's study shows it can be unstable. This was confirmed subsequently by several authors (Penven et al., 2014; Ioannou et al., 2019). In particular, the method is not suitable when the cyclogeostrophic equation has no solution. Further details are given in Sect. 2.

This paper proposes a new and modern numerical solution for the cyclogeostrophic inverse problem. The first novelty lies in its mathematical formulation as a minimization problem. The second novelty lies in the use of the JAX Python library to solve the optimization problem numerically. These developments make a new, open-source, and numerically efficient Python package for the cyclogeostrophic inversion, named jaxparrow. The minimization-based resolution corrects the shortcomings of the historical fixed-point method and enables a quantification of the impact of cyclogeostrophic corrections as effective resolution of SSH fields increases.

The paper is structured as follows: Sect. 2 reviews the analytic gradient wind solution and Arnason's fixed-point method for the cyclogeostrophic inversion, describes the new minimization-based method, and its implementation with JAX. Section 3 details the data used and the experimental setup of our study. Section 4 presents global applications with operational SSH maps: DUACS, available through the Copernicus Marine Environment Monitoring Service (CMEMS); and NeurOST, available through the Physical Oceanography Distributed Active Archive Center (PODAAC). Our proposed method is also compared to the fixed-point approach using pseudo-SWOT observations generated from the eNATL60 simulation. Finally, for both DUACS and NeurOST products, assessments of the derived currents using drifters are included.

## 2 Solutions to the cyclogeostrophic inversion problem

This section presents methods used to solve the cyclogeostrophic inversion problem. We first revisit the analytic gradient wind solution. We then review the historical fixed-point approach proposed by Arnason et al. (1962), which has been widely used despite known limitations. Finally, we introduce a novel minimization-based formulation of the inversion problem that addresses some of these shortcomings, and we describe our practical implementation of this minimization-based approach using modern automatic differentiation tools.

## 2.1 The analytic gradient wind solution

As discussed by Knox and Ohmann (2006), in an idealized circular and axisymmetric flow, the nonlinear term $(\boldsymbol{u}_{cg} \cdot \nabla)\boldsymbol{u}_{cg}$ simplifies to the centrifugal acceleration $-\frac{V_{gr}^2}{R}$ with $V_{gr}$ the azimuthal component of the velocity and $R$ the radius of curvature (which coincides with the radial distance to the vortex center in strictly axisymmetric cases). Under these assumptions, Eq. 3 becomes:

$$\frac{V_{gr}^2}{R} + fV_{gr} - fV_g = 0 \tag{4}$$

where $V_g$ is the azimuthal geostrophic velocity, positive for cyclonic eddies and negative for anticyclonic ones. Solving this quadratic equation yields the physically relevant branch of the gradient wind solution:

$$V_{gr} = \frac{2V_g}{1 + \sqrt{1 + 4V_g/(fR)}} \tag{5}$$

Equation 5 provides useful intuition about the conditions under which the cyclogeostrophic balance admits a physical solution. For cyclonic eddies ($V_g > 0$), the term under the square root is always positive, and a real solution exists. In contrast, for anticyclonic eddies ($V_g < 0$) this term becomes negative when $|V_g/(fR)| > 0.25$, in which case no real solution exists. This situation corresponds to the occurrence of inertial instability, indicating a breakdown of the balance assumptions (Knox and Ohmann, 2006).

## 2.2 State of the art: Arnason's (1962) fixed-point method

While the analytic gradient wind solution is useful for understanding the existence and physical limits of cyclogeostrophic balance, it is restricted to idealized axisymmetric flows. In realistic oceanic conditions—where the flow is neither perfectly circular nor steady—numerical approaches are instead required to solve Eq. 3. A widely used strategy is the fixed-point method originally proposed by Arnason et al. (1962), which we describe below.

Taking the vector product of $\mathbf{k}$ with Eq. 3 and substituting the geostrophic velocity $\boldsymbol{u}_g$ from Eq. 2, we obtain:

$$\boldsymbol{u}_{cg} - \frac{\boldsymbol{k}}{f} \wedge (\boldsymbol{u}_{cg} \cdot \nabla)\boldsymbol{u}_{cg} = \boldsymbol{u}_g \tag{6}$$

Then the iterations proposed by Arnason et al. (1962) to get the cyclogeostrophic velocity are initialized with $\boldsymbol{u}_{cg}^{(0)} = \boldsymbol{u}_g$ and implement as:

$$\boldsymbol{u}_{cg}^{(n+1)} = \boldsymbol{u}_g + \frac{\boldsymbol{k}}{f} \wedge (\boldsymbol{u}_{cg}^{(n)} \cdot \nabla)\boldsymbol{u}_{cg}^{(n)} \tag{7}$$

This approach has traditionally been referred to as the "iterative" method. However, this terminology can be misleading, as other numerical procedures—including our minimization-based formulation (see Sect. 2.3)—are also iterative while relying on fundamentally different update mechanisms. For clarity, we therefore adopt the more precise term "fixed-point" method to describe Eq. 7.

As initially mentioned by Arnason et al. (1962), these iterations do not always converge; an *ad hoc* and imperfect stopping strategy is generally implemented to avoid their numerical divergence. A typical case of numerical divergence is when the cyclogeostrophic equation has no solution, as previously discussed in Sect. 2.1. From a fixed-point perspective, divergence also occurs whenever the initial guess $\boldsymbol{u}_{cg}^{(0)} = \boldsymbol{u}_g$ is not an attracting fixed point of the update map in Eq. 7. Knox and Ohmann (2006) provide a detailed analysis of the convergence properties of this method in the context of the idealized gradient wind balance.

To mitigate these difficulties, Penven et al. (2014) stops the iterations at any grid point $i$ when the residual $|\boldsymbol{u}_{cg,i}^{(n+1)} - \boldsymbol{u}_{cg,i}^{(n)}|$ falls below 0.01 or starts to increase. Ioannou et al. (2019) implements this with two additional ingredients: the initial geostrophic velocity field is projected, with a cubic interpolation, on a grid 3 times finer than the initial one. This is to "improve the computation of the velocity derivatives" in Eq. 7. The second modification is in the calculation of the residual norm for each grid point, which includes now the 8 neighboring grid points.

Nonetheless, our own experience indicates that (i) the fixed-point method can fail to converge to the cyclogeostrophic solution (Fig. 1) and (ii) the local iteration-stopping strategy can produce noisy or unrealistic velocity fields (Fig. 2). These limitations motivate the need for an alternative approach.

## 2.3 Minimization-based formulation

We recast the cyclogeostrophic inversion problem in a minimization form, by searching for the velocity field $\boldsymbol{u}_{cg}$ that minimizes the following loss function:

$$J(\boldsymbol{u}_{cg}) = \int_{\Omega} [\Delta_{cg}(\boldsymbol{u}_{cg}(\boldsymbol{x}))]^2 d\boldsymbol{x} \tag{8}$$

where $\Omega$ is the 2D spatial domain and $\Delta_{cg}$ denotes the *cyclogeostrophic imbalance* function computed locally at each point $\boldsymbol{x} = (x, y)$ in the discretized domain:

$$\Delta_{cg}(\boldsymbol{u}_{cg}) = \left\| \boldsymbol{u}_{cg} - \frac{\boldsymbol{k}}{f} \wedge (\boldsymbol{u}_{cg} \cdot \nabla)\boldsymbol{u}_{cg} - \boldsymbol{u}_g \right\| \tag{9}$$

where $\|\cdot\|$ is the $\ell^2$ norm for a 2-component velocity vector: $\|\boldsymbol{u}(\boldsymbol{x})\| = \sqrt{u(\boldsymbol{x})^2 + v(\boldsymbol{x})^2}$, using the standard notation for the zonal and meridional velocities. In Eq. 8, we make it explicit that the loss function is the domain integral of a locally computed norm, although it could equivalently be expressed using an $L^2$ norm over the domain. This explicit form is useful for the discussion in Sect. 5.

The minimization of Eq. 8 is performed using gradient descent, i.e. by taking small steps in the direction opposite to the gradient of $J$:

$$\boldsymbol{u}_{cg}^{(n+1)} = \boldsymbol{u}_{cg}^{(n)} - \gamma \nabla J\left(\boldsymbol{u}_{cg}^{(n)}\right) \tag{10}$$

where the hyperparameter $\gamma$ controls the step size. The gradient $\nabla J\left(\boldsymbol{u}_{cg}^{(n)}\right)$ is computed automatically using JAX's reverse-mode automatic differentiation: JAX records the computation of $J$ as a sequence of elementary operations with known derivatives and applies the chain rule to construct the corresponding gradient function.

The minimization-based formulation is expected to solve the numerical divergence problem of the fixed-point method; it also provides a measure of the deviation from the cyclogeostrophic solution (when it exists). Where the cyclogeostrophic imbalance $\Delta_{cg}$ reaches 0, the solution is the cyclogeostrophic velocity. In regions where no exact cyclogeostrophic solution exists, the minimization-based approach—because its update strategy relies on the gradient of a globally evaluated loss involving spatial derivatives—favors a smoother and more coherent estimate of the velocity field, despite the absence of any explicit

regularization term. In this sense, it is expected to avoid the unrealistic features that the fixed-point method can generate, since the latter's point-wise update and stopping criterion tend to amplify noise. Interestingly, the cyclogeostrophic imbalance is a straightforward indication of where a cyclogeostrophic velocity can be found, and where it cannot. It is not possible to determine the physical nature of the velocity solution when $\Delta_{cg}$ does not reach 0. But it is still possible to quantify a deviation from the cyclogeostrophic equilibrium.

### 2.4 Implementation

Our cyclogeostrophic inversion library, `jaxparrow` (Bertrand et al., 2025), is implemented with `JAX` (Bradbury et al., 2018), a Python library developed by Google to perform two main operations on Python functions: acceleration and automatic differentiation. `jaxparrow` leverages both features. The automatic differentiation capability directly provides the gradient of $J$, which can be used for gradient-based minimization methods. For the minimization itself, `jaxparrow` implements `Optax`

(DeepMind et al., 2020), a gradient processing and optimization library specifically developed for `JAX`.

jaxparrow handles gridded data, making it well-suited for estimating cyclogeostrophic currents from SSH derived from models, Level-4 products, and also 2D Level-3 products. While most altimetry products use Arakawa A-grids, where all quantities are evaluated at the grid center (T point), `jaxparrow` computes partial derivatives using finite differences on Arakawa C-grids, where the SSH is defined at the grid center, the velocity components at the grid faces, and the vorticity at the

165 grid vertices. As a result, variables must be carefully interpolated when performing numerical computations. Specifically, for the kinematic diagnostics described in Sect. 3.2.1, the velocity components $u$ and $v$ are first interpolated to the T points prior to computing the velocity magnitude, whereas vorticity is calculated directly on the C-grid and then interpolated back to the T points.

To support further evaluation of our minimization-based method and facilitate the integration of cyclogeostrophic currents

into a global operational product, our library is easily installable via `pip`, with its code publicly available on GitHub.

## 3 Data and experimental setup

This section describes the data sources and methodology used to assess cyclogeostrophic surface current reconstructions. We first present the satellite-derived products, the model data, and the drifter dataset used for validation. We then detail the experimental setup, including the computation of derived kinematic fields and the evaluation procedure based on drifter-derived velocities.

## 3.1 Input and validation data

### 3.1.1 Operational SSH products

Following Penven et al. (2014), Ioannou et al. (2019), and Cao et al. (2023), we use the standard Data Unification and Altimeter Combination System (DUACS, 2024) SSH global product. As reported by Ballarotta et al. (2019), the DUACS effective resolution (computed using the Signal to Noise Ratio method) ranges globally from 100 km at high latitudes to 800 km in the equatorial band. In its most recent version, DUACS provides data at daily temporal increments on a 1/8° spatial grid.

To illustrate the relevance of cyclogeostrophic corrections as effective resolution increases, we also use the newer experimental global product NeurOST (NeurOST, 2024). NeurOST gridded data have a temporal resolution of one day and a spatial resolution of 1/10°. Martin et al. (2024) shows that by combining satellite observations of SSH and SST, NeurOST improves the effective resolution by up to 30 % compared to DUACS, particularly in the Gulf Stream region, where NeurOST achieves an effective resolution of 108 km versus 150 km for DUACS.

The present study covers the period from 2010 to 2022 (inclusive), corresponding to the availability period of both DUACS and NeurOST products.

### 3.1.2 eNATL60 model data

We leveraged SSH and surface currents from the eNATL60-BLB002 simulation (Brodeau et al., 2020) to illustrate the benefits of reconstructing surface currents from SSH using the cyclogeostrophic approximation rather than the geostrophic one. eNATL60 is a submesoscale-permitting North Atlantic configuration (including the Mediterranean Sea) of the NEMO ocean model, with a 1/60° horizontal resolution. We employed the tide-free version of the configuration and the daily-averaged dataset of the simulation run.

### 3.1.3 Global Drifter Program (GDP) dataset

We used 6-hourly interpolated surface current velocity measurements from drifters, collected in the GDP database (Lumpkin and Centurioni, 2019). The GDP database includes data from drifters of various types and shapes with differing sensitivities to wind. To ensure that the reference velocities are not influenced by direct wind forcing on the drifters, we restricted our analysis to drogued SVP-type drifters. Drogue-loss detection in SVP drifters was known to be unreliable, leading to some observations being incorrectly tagged as drogued. The GDP database provides a more robust drogue presence tag, employing the procedure described by Lumpkin et al. (2013), in which drogue loss is detected based on anomalous downwind ageostrophic motion. At the global scale, over the period 2010–2022, it represents approximately 9.8 million observations from around 12,500 drifters.

### 3.1.4 Modeled Ekman currents

To remove the Ekman contribution from the drifter-derived velocities, we used the GlobCurrent product (GlobCurrent, 2024). In GlobCurrent, Ekman currents at the surface and at 15 m depth are estimated from ERA5 wind stress following the methodology of Rio et al. (2014). These estimates are provided at hourly resolution on a regular 1/4° grid.

## 3.2 Experimental setup

### 3.2.1 Derived kinematics

Starting from global SSH maps, we present several diagnostics to assess the impact of accurately computed cyclogeostrophic velocities.

We compute the cyclogeostrophic imbalance from Eq. 9 and use it as a local measure of deviation from cyclogeostrophy, expressed in $m\,s^{-1}$. To better highlight divergences while estimating cyclogeostrophic currents, spatial deviations from cyclogeostrophy are aggregated over time by taking the maximum of the 7-day moving average, following the approach of Fig. 12 in Ioannou et al. (2019).

We derive geostrophic and cyclogeostrophic velocities from SSH using `jaxparrow`. Fixed-point cyclogeostrophic velocities are computed using Eq. 7, with the stopping procedure described in Sect. 2.2 (same as Penven et al., 2014; Cao et al., 2023). Minimization-based cyclogeostrophic velocities are estimated by minimizing $J$ (Eq. 8) using gradient descent (Eq. 10) with a fixed step size of $5 \times 10^{-3}$ for 2,000 iterations, using geostrophic velocities as the initial guess.

Relative vorticity provides insight into ocean dynamics and the quality of reconstructed current velocities. It represents the spinning motion of a water parcel relative to the Earth and is defined as the curl of the velocity:

$$\zeta = \frac{\partial v}{\partial x} - \frac{\partial u}{\partial y} \tag{11}$$

Since it requires computing spatial derivatives, it is expected to highlight noise in velocity fields. Relative vorticity maps are also computed using `jaxparrow`.

Eddy Kinetic Energy (EKE) quantifies the kinetic energy associated with the time-varying component of the flow and as such is a good indicator of the mesoscale dynamics. Following Cao et al. (2023) we compute it as:

$$\text{EKE} = \frac{(u')^2 + (v')^2}{2} \tag{12}$$

where $u'$ and $v'$ are the zonal and meridional components of the velocity anomaly (i.e. deviation from the mean flow). We use the Sea Surface Height Anomaly (SSHA), rather than the full SSH, to compute geostrophic and cyclogeostrophic velocity anomalies in the same manner as for total current velocity.

### 3.2.2 Evaluation against total surface currents

To validate that cyclogeostrophy provides a better estimate of surface currents than geostrophy, we compute evaluation metrics against eNATL60 relative vorticity and drifter-derived velocities.

**Pseudo-SWOT observations of the eNATL60 SSH**

Because the true total sea-surface fields corresponding to satellite SSH observations are unknown, one way to evaluate the cyclogeostrophic inversion methods is the use of model data. To mimic SWOT swath observations from model output, we generate pseudo-SWOT data by re-interpolating eNATL60 SSH onto portions of the two SWOT CalVal passes that cross the Balearic Sea, using the 2-km SWOT grid. For the purpose of showcasing the minimization-based method and comparing it to the fixed-point approach in a controlled setting, we did not add artificial noise to eNATL60 SSH. Consequently, the pseudo-SWOT data used here do not include the measurement and geophysical errors affecting real SWOT observations, which are discussed extensively in the literature (e.g. Nencioli et al., 2025; Peral et al., 2024; Wang et al., 2025).

For each point of the SWOT grid, we define the inversion error for method $M$ as:

$$\epsilon_M = (\zeta_M - \zeta)^2 \tag{13}$$

where $\zeta$ and $\zeta_M$ are relative vorticity fields computed from Eq. 11 using, respectively, the eNATL60 velocity field (interpolated onto the SWOT swath) and the velocity field obtained from the cyclogeostrophic ($M = cg$) or geostrophic ($M = g$) inversion of the eNATL60 SSH field, also interpolated onto the swath. We then compute the time-averaged Root Mean Square Error (RMSE) at each grid point over August 2009:

$$\mathrm{RMSE}_M = \sqrt{\frac{1}{N} \sum_{i=1}^{N} \epsilon_M^{(i)}} \tag{14}$$

where $N = 31$ is the number of days considered. To compare two inversion methods, we use the normalized difference between their RMSE values:

$$\Delta\,\mathrm{RMSE}_{M_1 - M_2} = 100 \frac{\mathrm{RMSE}_{M_1} - \mathrm{RMSE}_{M_2}}{\mathrm{RMSE}_{M_1}} \tag{15}$$

This indicator measures the relative improvement (or degradation) in the fidelity of the reconstructed vorticity field when using method $M_2$ instead of $M_1$, capturing changes in both bias and variance of the inversion.

**Velocities derived from drogued SVP drifters**

Another way to evaluate the cyclogeostrophic inversion methods is to use drifter-derived velocities.

Thanks to their drogue centered at 15 m depth, SVP drifters sample the currents in the upper $\sim$10–20 m of the ocean (Lumpkin and Pazos, 2007). They provide an estimate of the total current velocity, including signatures from high-frequency processes such as near-inertial wave, and, as illustrated by Eq. 1, these unbalanced motions are neglected in both the geostrophic and the cyclogeostrophic approximations. To mitigate the influence of these additional terms in our analysis, we follow the procedure applied by Müller et al. (2019) to 6–hourly interpolated SVP drifter data. We first remove the Ekman contribution to the drifter velocities using the 15 m Ekman current estimated in the GlobCurrent product. We then filter near-inertial signal by applying a second-order Butterworth filter with a 25–hour cutoff period to the drifter velocities.

For each drifter observation $i$ at time $t_i$ and position $\mathbf{X}_i$, we define the inversion error for method $M$ as:

$$\epsilon_M^{(i)} = \|\mathbf{u}_M(t_i, \mathbf{X}_i) - \mathbf{u}_d^{(i)}\|^2 \tag{16}$$

where $\mathbf{u}_d$ is the drifter velocity vector and $\mathbf{u}_M$ is the velocity field obtained from the cyclogeostrophic ($M = cg$) or geostrophic ($M = g$) inversion, interpolated at the drifter time and position. Individual errors are binned into 1° latitude × 1° longitude boxes (Fig. B1 shows the number of observations per bin). Within each bin, we compute the RMSE of an inversion method $M$ using Eq. 14, where $N$ is now the number of errors (or observations) inside that bin. To compare two inversion methods spatially, we use the normalized difference between their binned RMSE values, as defined in Eq. 15.

In addition to the spatial comparison, we also assess whether the cyclogeostrophic solution provides a better estimate than the geostrophic one as a function of the magnitude of the cyclostrophic correction. The cyclostrophic correction is defined as the difference between the cyclogeostrophic velocity (obtained using the minimization-based approach) and the geostrophic velocity:

$$\mathbf{u}_c = \mathbf{u}_{cg} - \mathbf{u}_g \tag{17}$$

For any drifter observation $i$, we consider the cyclogeostrophic solution to be better than the geostrophic one if $\epsilon_{cg}^{(i)} < \epsilon_g^{(i)}$. This criterion allows us to model the probability that the cyclogeostrophic solution outperforms the geostrophic one, conditionally on the magnitude of the cyclostrophic correction, $\mathbb{P}(\epsilon_{cg} < \epsilon_g \|\|\mathbf{u}_c\|)$, using a logistic regression. To allow for a nonlinear dependence on $\|\mathbf{u}_c\|$, we expand $x_i = \|\mathbf{u}_c^{(i)}\|$ using a natural cubic spline basis $\mathbf{s}(x_i) = \big(s_1(x_i), \ldots, s_K(x_i)\big)^\top$ with $K = 4$ functions, and fit the model:

$$\text{logit}[p_i] = \beta_0 + \sum_{k=1}^{K} \beta_k s_k(x_i), \qquad \text{with} \qquad p_i = \mathbb{P}\left(\epsilon_{cg}^{(i)} < \epsilon_g^{(i)} \|\|\mathbf{u}_c^{(i)}\|\right). \tag{18}$$

This provides a smooth estimate of the probability that cyclogeostrophy outperforms geostrophy as a function of the cyclostrophic correction magnitude, along with 95% confidence bands computed using the delta method.

## 4  Application to SSH maps

In this section, we apply the proposed cyclogeostrophic inversion method to maps of SSH. We first analyze the resulting geostrophic and cyclogeostrophic surface currents at the global scale, highlighting differences in dynamic regions. We then focus on the reconstruction skill using pseudo-SWOT swath observations over the Balearic Sea. Finally, we evaluate the reconstructed currents globally by comparing them with independent drifter measurements from the GDP. Unless otherwise specified, the cyclogeostrophic inversion method referred to throughout this section is the minimization-based one.

### 4.1  Analysis of geostrophic and cyclogeostrophic currents

Surface currents derived from SSH using the geostrophic approximation and both minimization-based and fixed-point cyclogeostrophic inversion methods are here qualitatively analyzed (i) with the cyclogeostrophic imbalance from Eq. 9, (ii) by

290 observing the velocity and relative vorticity fields, and (iii) through a comparison of EKE.

  The measure of the deviation from cyclogeostrophy shows that (i) geostrophy can be a crude approximation of cyclogeostrophy at some locations in space and time and (ii) the minimization-based inversion method is more accurate than the fixed-point method to compute a cyclogeostrophic velocity field. These conclusions are drawn from the examination of Fig. 1 which

295 presents the time-aggregated deviation from the cyclogeostrophic balance of 3 velocity fields derived from NeurOST SSH, namely the geostrophic field (top) and the cyclogeostrophic solutions from the minimization-based method (bottom left) and the fixed-point method (bottom right). The geostrophic field exhibits large deviations from cyclogeostrophy, with deviations larger than 0.3 m s$^{-1}$ at nearly 5 % of grid points, hinting that the advective term should not be neglected. The solution of the fixed-point method deviates even further, with differences exceeding 0.35 m s$^{-1}$ at more than 5 % of points. In contrast,

300 the minimization-based method limits deviations above 0.03 m s$^{-1}$ to fewer than 5 % of grid points. This suggests that the fixed-point method is less reliable in converging toward a cyclogeostrophic solution, particularly in the western boundary currents, where the minimization-based method shows that a cyclogeostrophic solution exists.

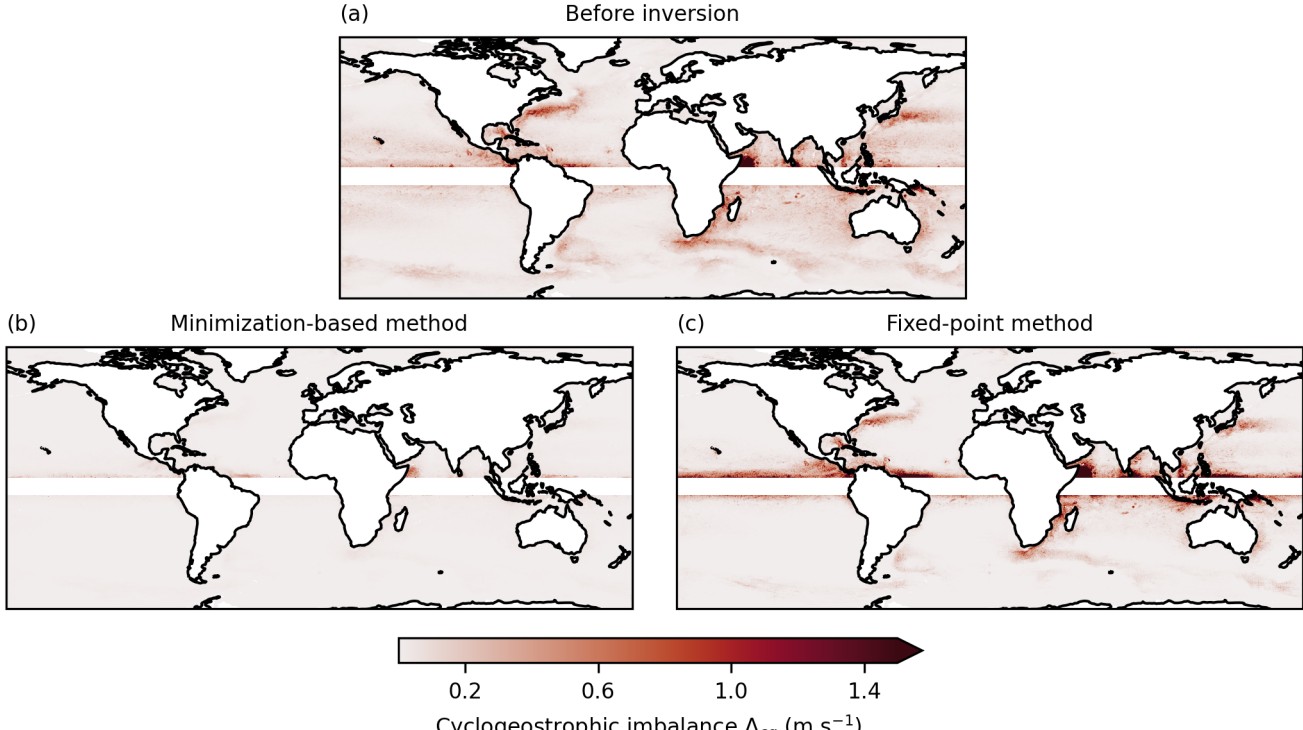

**Figure 1.** Maps of cyclogeostrophic imbalance, computed from Eq. 9, for the geostrophic velocity (a), the minimization-based cyclogeostrophic velocity (b), and the fixed-point cyclogeostrophic velocity (c) derived from NeurOST SSH.

Our implementation of the proposed minimization-based method enables physically consistent estimation of cyclogeostrophic currents on a global scale, including in highly dynamic regions where cyclogeostrophic corrections substantially impact jets and eddies, and where the fixed-point method yields unrealistic physical fields. Figure 2 presents a global snapshot of the norm of cyclogeostrophic currents derived from NeurOST SSH, along with an enlargement of the Gulf Stream region where relative vorticity and differences compared to geostrophy are also displayed. In the northern meanders of the Gulf Stream jet, cyclogeostrophic corrections are positive and can reach up to $+0.2$ m s$^{-1}$, while in the southern meanders they are negative, down to $-0.2$ m s$^{-1}$. Similarly, anticyclonic and cyclonic eddies exhibit respective cyclogeostrophic contributions of approximately $+0.2$ m s$^{-1}$ and $-0.2$ m s$^{-1}$, corresponding to relative increases of 10–50 % in the anticyclonic case and relative decreases of 10–50 % in the cyclonic case. Finally, while the minimization-based method allows for the reconstruction of a smooth and physically coherent relative vorticity field, the fixed-point method introduces artifacts in the most dynamic parts of the jet and eddies. As discussed in Sections 2.2 and 2.3, the differences are likely linked to the mathematical distinctions between the two approaches.

The EKE computed from the geostrophic and the minimization-based cyclogeostrophic velocities anomalies exhibit differences up to 20 %, essentially at low and middle latitudes. This is shown in Fig. 3, which presents the relative difference in EKE between cyclogeostrophy and geostrophy, averaged over the whole time period. Positive differences are particularly pronounced near the equatorial band. Regions with intense dynamics such as the western boundary currents are characterized by elongated dipole structures with both positive and negative differences. These reflect a current intensification in anticyclonic eddies detaching poleward and a damping of the current in cyclonic eddies detaching equatorward, in agreement with the magnitude and sign of cyclogeostrophic corrections observed in Fig. 2. All these observations are consistent with Cao et al. (2023) who performed a similar analysis with 1/4° DUACS maps and the historical fixed-point method for cyclogeostrophy over the period 1993-2018. Our results suggest once more that geostrophy can be a crude approximation leading to errors up to 20 % in EKE.

## 4.2 Evaluation using pseudo-SWOT observations from eNATL60

Normalized relative vorticity fields obtained from geostrophy or cyclogeostrophy surface current reconstruction are compared to reference fields derived from eNATL60 total surface currents. To demonstrate the feasibility of performing the cyclogeostrophic inversion in the SWOT swath using our package `jaxparrow`, the original eNATL60 fields are first interpolated onto the 2-km grid of the SWOT swath before reconstruction.

While normalized relative vorticity fields derived from the cyclogeostrophic balance are generally in better agreement with eNATL60 reference—especially near the cores of persistent anticyclonic eddies—the fixed-point method more frequently exhibits RMSE increases compared to geostrophy, particularly along eddies boundaries where the minimization-based approach continues to outperform geostrophy. This is illustrated in Fig. 4, which shows a snapshot of the reference normalized relative vorticity field (top-left), the RMSE of the minimization-based reconstruction computed over one month (top-right), and the relative change in RMSE with respect to geostrophy for both the fixed-point method (bottom-left) and the minimization-based method (bottom-right). Figure A1 also displays the normalized relative vorticity field for the three inversion methods, together with the corresponding surface current velocity fields. Several anticyclonic submesoscale ($\leq$50 km) eddies can be identified in

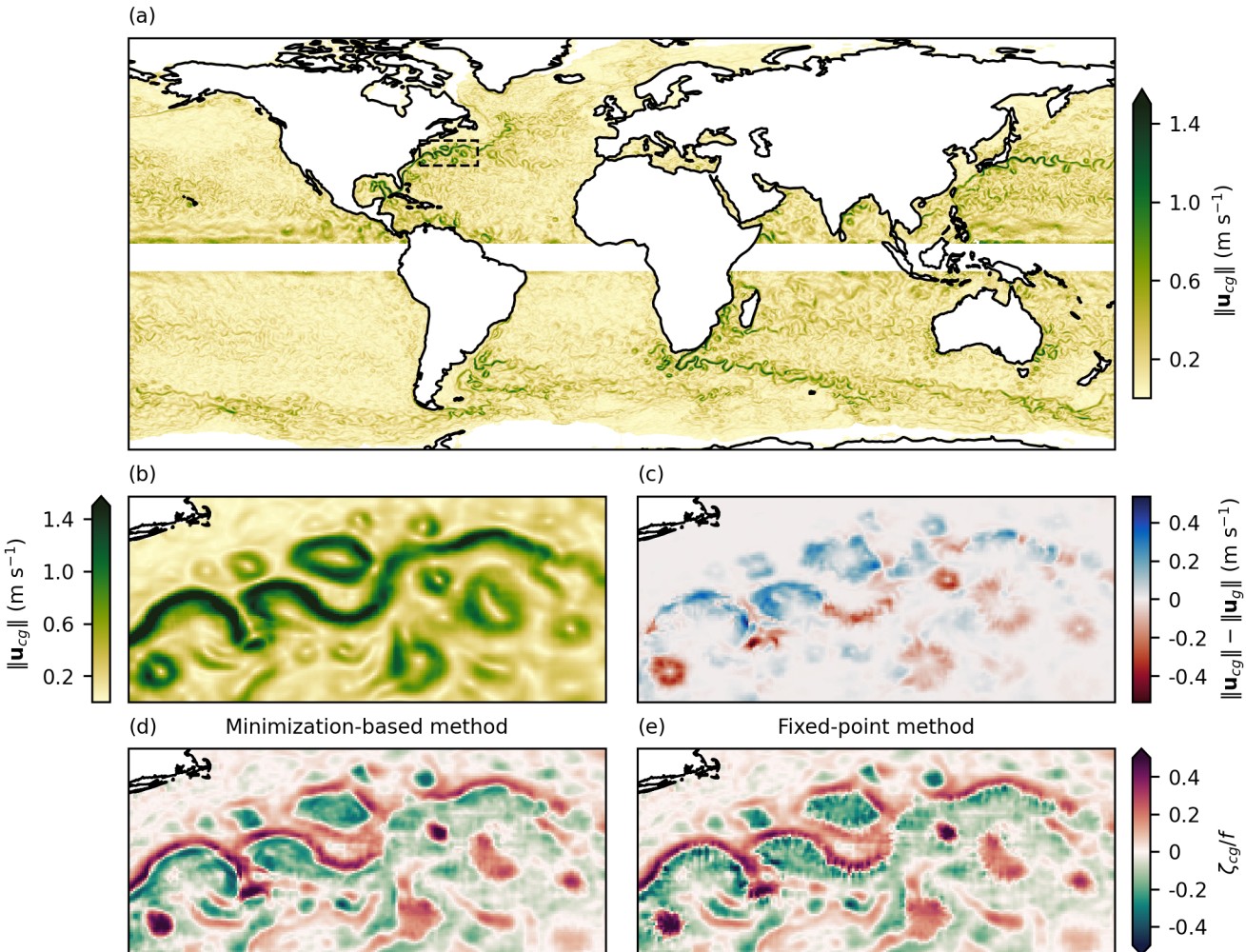

**Figure 2.** 16 April 2015 snapshots derived from NeurOST SSH. (a) Norm of the minimization-based cyclogeostrophic velocity. (b) Same as (a), zoomed in the Gulf Stream region. (c) Difference between the norms of minimization-based cyclogeostrophic and geostrophic velocities. (d) Relative vorticity computed from the minimization-based cyclogeostrophic velocity. (e) Same as (d), using the fixed-point cyclogeostrophic velocity.

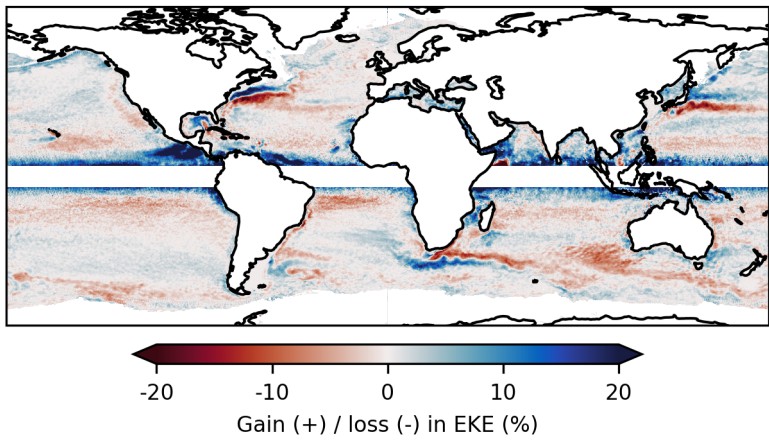

**Figure 3.** Relative difference in EKE between minimization-based cyclogeostrophic and geostrophic current velocity anomalies derived from NeurOST SSH.

the reference normalized relative vorticity field shown in panel (a). Three of these eddies—located North and South of Ibiza, and South of Menorca—are persistent over the full month of the evaluation period (not shown). From panel (b), we observe that the RMSE of the minimization-based method exceeds 0.1 only in coastal areas, where the cyclogeostrophic assumption likely breaks down. The relative difference in RMSE with respect to geostrophy in panel (d) generally indicates a better reconstruction when using the minimization-based approach, particularly in the regions of the three persistent eddies where improvements locally reach 100 %. Conversely, panel (c) shows that the fixed-point method provides slightly weaker improvements and, more notably, more frequent degradations, with pixel-like patterns similar to the artifacts seen in Fig. 2 and also noticeable in Fig. A1.

Consistently with Archer et al. (2025) and Tranchant et al. (2025), these results suggest that cyclogeostrophy should be employed when analyzing high-resolution 2D SSH fields. They also indicate that the minimization-based method may provide more reliable reconstructions than the fixed-based approach in such contexts.

### 4.3 Evaluation with data from the GDP

Reconstructed cyclogeostrophic and geostrophic currents are evaluated against drifter-derived velocities using (i) the inversion error defined in Eq. 16 and binned within $1°$ latitude $\times$ $1°$ longitude boxes at the global scale, and (ii) a logistic regression modeling the probability for cyclogeostrophy to outperform geostrophy as a function of the cyclostrophic correction magnitude.

When using NeurOST SSH, minimization-based cyclogeostrophic corrections improve surface current estimates, particularly in energetic regions such as western boundary currents, where reconstruction errors are highest. This is illustrated in Figures 5 and 6. Figure 5 presents global maps of the cyclogeostrophic RMSE obtained from NeurOST SSH (top-left panel) and of the comparison between cyclogeostrophic and geostrophic inversion methods for NeurOST (top-right). Cyclogeostrophic RMSE remains below 0.1 m s$^{-1}$ across most of the ocean but increases to 0.2–0.5 m s$^{-1}$ in energetic currents. In these regions,

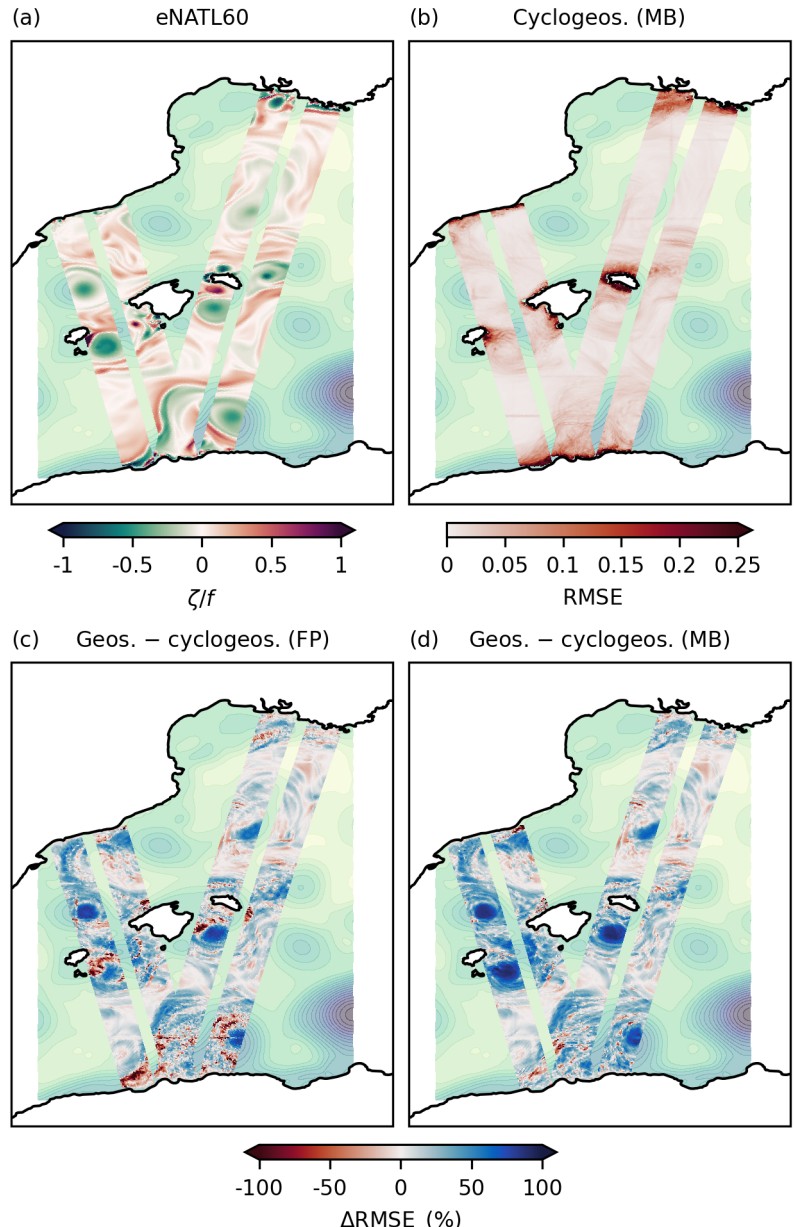

**Figure 4.** Performance of cyclogeostrophic inversion methods applied to eNATL60 SSH interpolated onto the SWOT swath. The background field in all four panels is the original eNATL60 SSH on 15 August 2009. (a) Normalized relative vorticity computed from eNATL60 surface currents on 15 August 2009. (b) RMSE obtained when reconstructing surface currents using the minimization-based approach. (c) Relative RMSE difference between geostrophic and fixed-point cyclogeostrophic inversions. (d) Same as (c) but using the mimization-based cyclogeostrophic inversion. RMSE values of normalized relative vorticity with respect to eNATL60 in panels (b), (c) and (d) are computed over the full month of August 2009.

NeurOST-based cyclogeostrophy clearly reduces error, with improvements of up to 10 % in the Gulf Stream and over 20 % in the Kuroshio (see the insets in the top-right panel of Fig. 5, which highlight the error reductions in these western boundary currents). Figure 6 further illustrates this, showing the probability that cyclogeostrophy outperforms geostrophy as a function of the

360 cyclostrophic correction magnitude. The solid lines correspond to the logistic regression fit, and the shaded envelopes indicate the 95 % confidence bands. We note that these confidence bands are estimated from the whole population of inversion errors, that is why the binned empirical mean probabilities (dots)—which are computed from smaller subsets of data as the cyclostrophic corrections increases—fall outside the bands. Focusing on NeurOST-derived currents (blue), we find that cyclogeostrophy is, on average, consistently a better estimate than geostrophy, and that this probability increases with the magnitude of the

365 cyclostrophic correction, up to 70 % for cyclostrophic corrections of $0.45 \ \mathrm{m \ s^{-1}}$.

In contrast, cyclogeostrophic corrections can degrade performances when applied to DUACS SSH. This is again illustrated in Figures 5 and 6. The bottom-left panel of Fig. 5 compares cyclogeostrophic and geostrophic inversion methods based on DUACS SSH. Unlike results obtained with NeurOST, regions such as the western boundary currents show a degradation in performance of around 10 % when cyclogeostrophic corrections are applied (see the insets in the bottom-left panel of Fig.

5, which highlight the increased error in these regions). Similarly, the orange line in Fig. 6 shows the logistic regression fit for improving the reconstruction when using cyclogeostrophy rather than geostrophy for DUACS-based surface currents. Cyclogeostrophy performs worse more often, on average, than geostrophy for cyclostrophic corrections smaller than $0.45 \ \mathrm{m \ s^{-1}}$. These discrepancies could stem from differences in the effective resolution of the SSH products: DUACS may insufficiently capture fine-scale structures, deteriorating the accuracy of cyclogeostrophic corrections in energetic regions.

Importantly, the combination of higher effective resolution SSH fields and cyclogeostrophic inversion yields substantial benefits over the current operational standard. As shown in Fig. 5 (bottom-right panel), applying minimization-based cyclogeostrophy to NeurOST SSH reduces reconstruction error by 5–20 % at mid-latitudes relative to DUACS geostrophy.

These results suggest that cyclogeostrophic corrections will become increasingly relevant as SSH products achieve higher effective resolution—consistent with the findings from Tranchant et al. (2025)—and could significantly benefit future operational

surface current products.

## 5 Discussion and conclusions

We developed a new and robust method for the cyclogeostrophic inversion of surface currents by reformulating the inversion problem in a minimization-based framework, thereby overcoming the limitations of the traditional fixed-point approach. The method is implemented as an open-source Python package, `jaxparrow`, which leverages the `JAX` library for high-performance

and scalable computation, enabling its application at the global scale. When applied to NeurOST SSH fields and pseudo-SWOT observations, the proposed approach yields physically consistent cyclogeostrophic current estimates, particularly in energetic regions. The relevance of the cyclogeostrophic corrections derived with our minimization-based method is supported by a global, 13-year comparison with drifter-derived velocities.

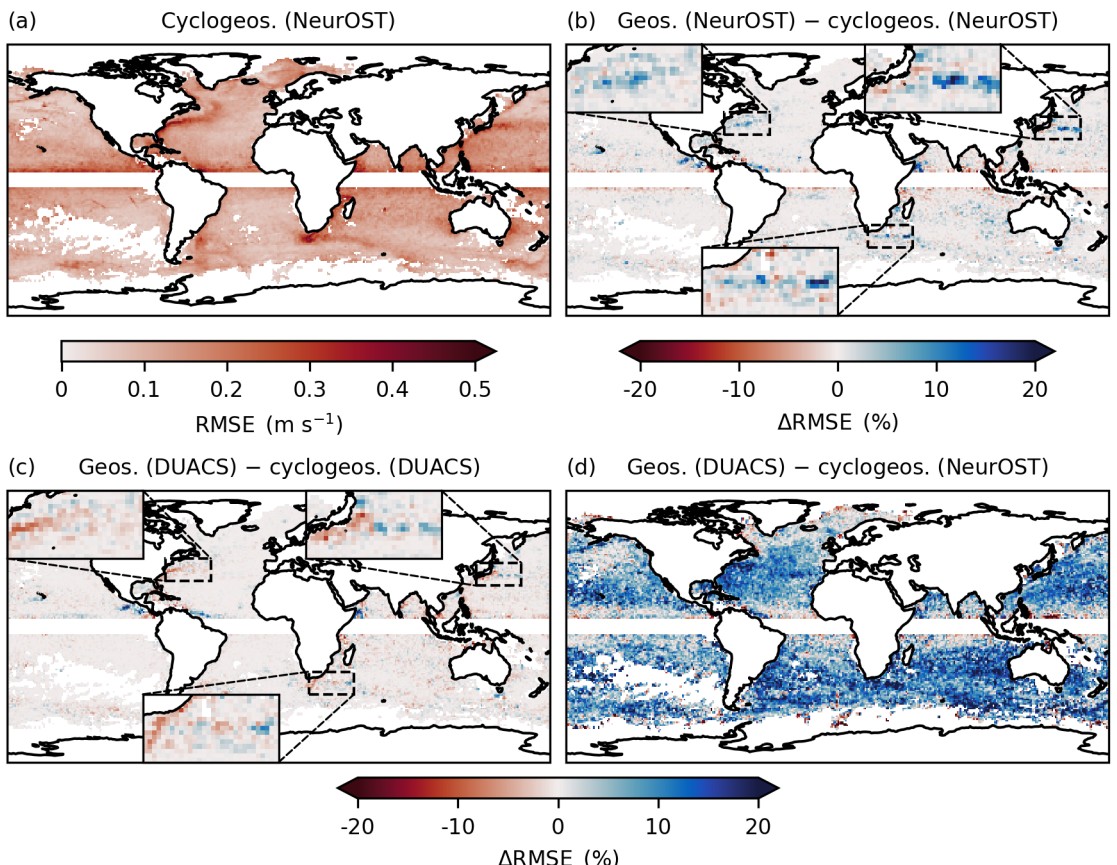

**Figure 5.** (a) RMSE with respect to the drifters for the cyclogeostrophic velocity estimated from NeurOST SSH. (b) Relative RMSE difference of NeurOST-derived geostrophic and cyclogeostrophic velocities. (c) Same as (b) but using SSH from DUACS. (d) Same as (b) but between DUACS geostrophic velocities and NeurOST cyclogeostrophic velocities.

This work makes systematic application of cyclogeostrophic inversion feasible, providing a complementary tool for recon-
390 structing surface currents from operational SSH products as well as from high-resolution 2D SSH observations in the SWOT swath.

Several questions were not addressed in this study. By formulating the cost functional $J$ from Eq. 8 as a domain integral, the solution to the minimization problem depends on the entire study region. Moreover, we did not investigate the sensitivity of the minimization solution to the choice of the optimizer: although Eq. 10 illustrates the classical gradient descent update, the `Optax`
library provides many alternative optimization algorithms and corresponding hyperparameters. These points suggest potential avenues for investigation, such as partitioning the domain into sub-regions and applying different minimization strategies tailored to the energetic conditions of each area. Furthermore, the iterations from Equations 7 and 10 are initialized using the geostrophic velocity field. An alternative—of potential interest for future work—would be to initialize from the analytical gradient wind

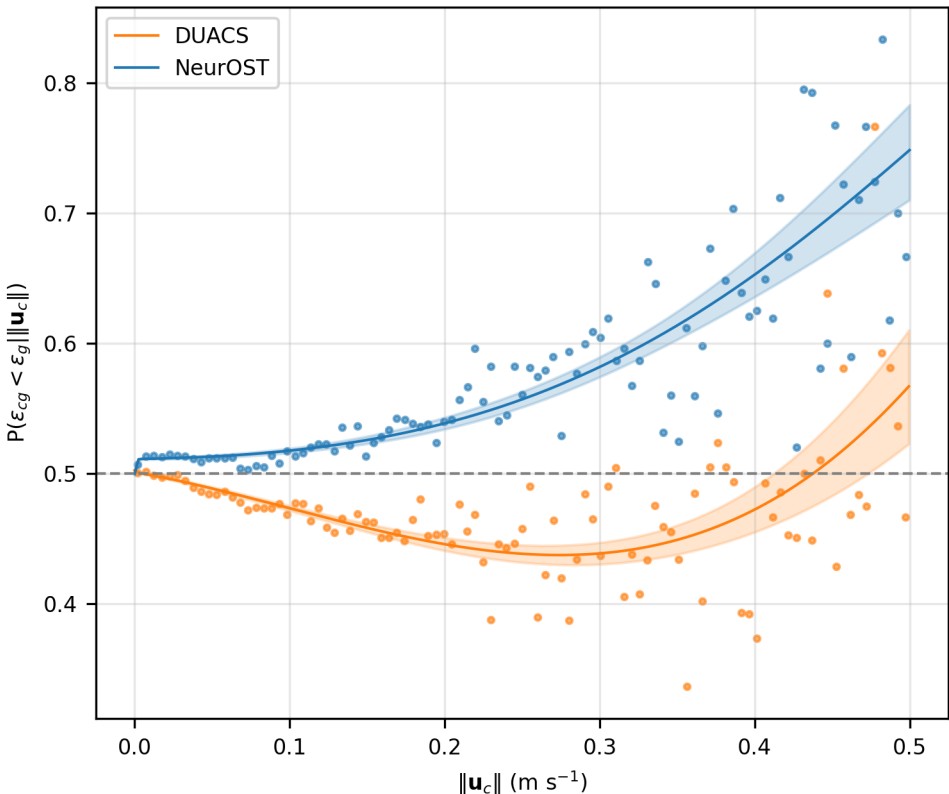

**Figure 6.** Probability that cyclogeostrophy improves surface current reconstruction relative to geostrophy, as a function of the cyclostrophic correction magnitude. Dots indicate empirical proportions computed per bin of cyclostrophic correction magnitude. Solid lines show the logistic regression fit. Shaded envelopes denote to the 95 % confidence band, computed using the delta method.

solution (Eq. 5), relaxing the axisymmetric assumption by estimating the local radius of curvature following Meijer et al. (2022)
(see their Eq. 3).

In addition to enabling the inclusion of cyclogeostrophic corrections in operational SSH and surface current products, our work opens several additional opportunities. With its effective resolution reaching 15 km within the swath, the SWOT mission offers unprecedented possibilities for observing and studying the submesoscales. While several efforts are currently underway to accurately separate balanced and unbalanced signals from SWOT SSH (Gao et al., 2024; Tranchant et al., 2025; Uchida
et al., 2025), our implementation provides a practical approach for reconstructing cyclogeostrophic currents from balanced SSH, thereby enabling SSH-based diagnostics to be systematically extended beyond the geostrophic approximation. Another advantage of our minimization-based formulation is its flexibility to incorporate extra constraints or regularization terms directly into the inversion. Because the cyclogeostrophic inversion is expressed as a differentiable cost functional, the method can naturally be extended to jointly filter noisy SSH observations—such as those from SWOT, similarly to Tranchant et al.
(2025)—or to enforce consistency with ancillary surface fields, like sea surface temperature as in Le Guillou et al. (2025).

While these extensions could also be embedded within larger variational or learning-based data-assimilation systems, the key advantage here is the ability to constrain the inversion itself using additional physical or observational information.

*Code and data availability.* The DUACS delayed-time altimeter gridded maps of sea surface height product used in this study is freely available on the CMEMS portal: https://doi.org/10.48670/moi-00148.

The NeurOST delayed-time altimeter gridded maps of sea surface height product used in this study is freely available on the PO.DAAC portal: https://doi.org/10.5067/NEURO-STV24.

The six-hourly interpolated drifters data used in this study is freely available on the NOAA portal: https://doi.org/10.25921/7ntx-z961, or via the `clouddrift` Python library: https://doi.org/10.5281/zenodo.11081647.

The SWOT L3 Expert data in its version v2_0_1 is available through the AVISO+ portal: https://doi.org/10.24400/527896/A01-2023.018.

The eNATL60-BL002 data is available on MEOM's OpeNDAP: https://ige-meom-opendap.univ-grenoble-alpes.fr/thredds/catalog/meomopendap/extract/MEOM/eNATL60/eNATL60-BLB002/1d/SSH/catalog.html.

The minimal diagnostics used in this study are available on Zenodo: https://doi.org/10.5281/zenodo.16099419. More comprehensive and larger datasets can also be found on MEOM's OpeNDAP: https://ige-meom-opendap.univ-grenoble-alpes.fr/thredds/catalog/meomopendap/extract/MEOM/cyclogeostrophy-paper/catalog.html.

The code used to run this study experiments and produce the diagnostics presented here can be found on GitHub: https://github.com/vadmbertr/cyclogeostrophy_impact_experiment.

The code of the Python library `jaxparrow` introduced in this paper is also available on GitHub: https://github.com/meom-group/jaxparrow.

## Appendix A:  Cyclogeostrophic inversion in a pseudo-SWOT swath

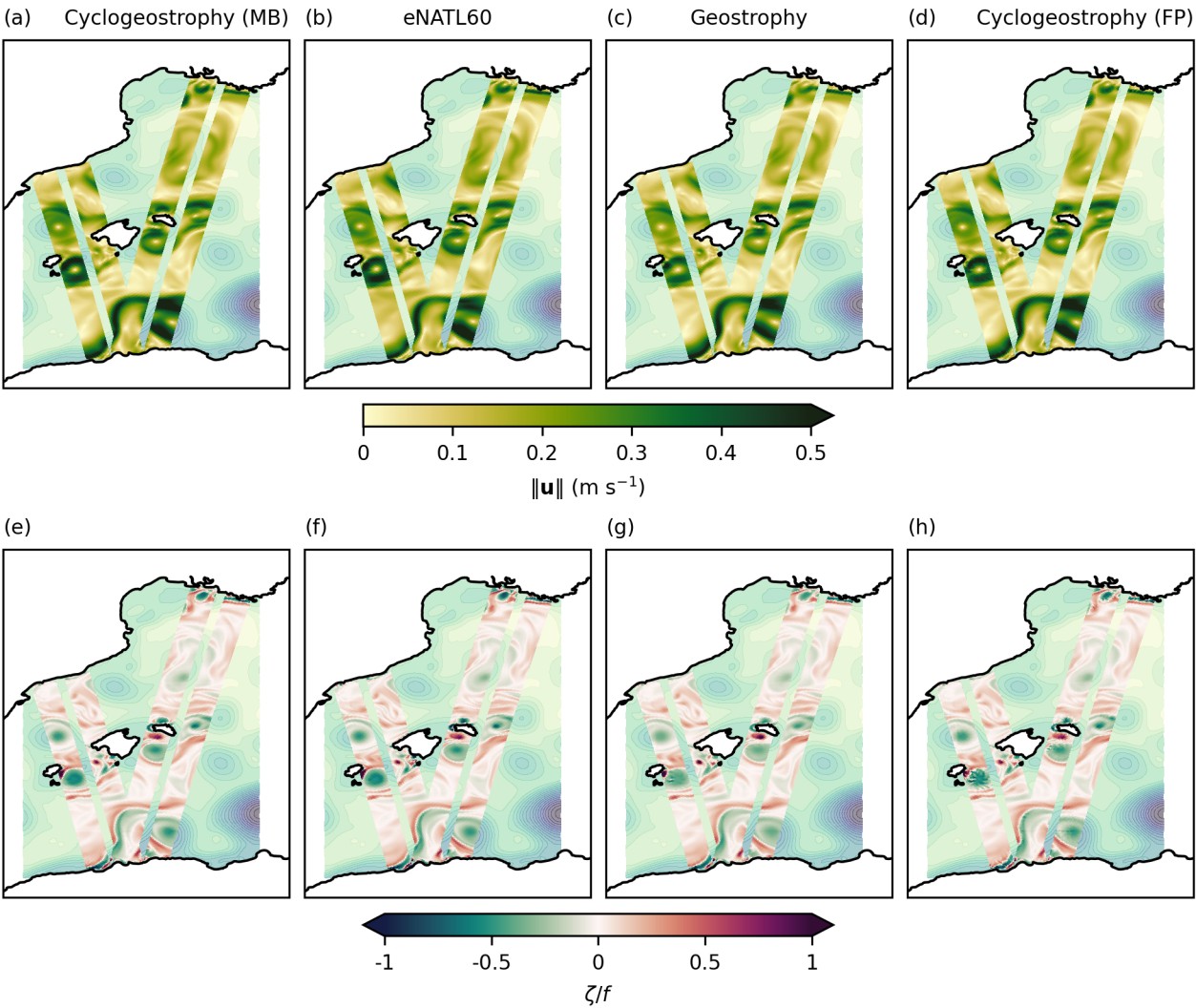

**Figure A1.** 15 August 2009 snapshots derived from eNATL60 SSH (background), interpolated onto the 2-km SWOT swath grid. Top row: surface current magnitude. Bottom row: normalized relative vorticity. (a), (e) Cyclogeostrophic currents reconstructed with the minimization-based method. (b), (f) True eNATL60 fields interpolated onto the swath. (c), (g) Geostrophic currents reconstructed from SSH. (d), (h) Cyclogeostrophic currents reconstructed with the fixed-point method.

## Appendix B: Evaluation with data from the GDP

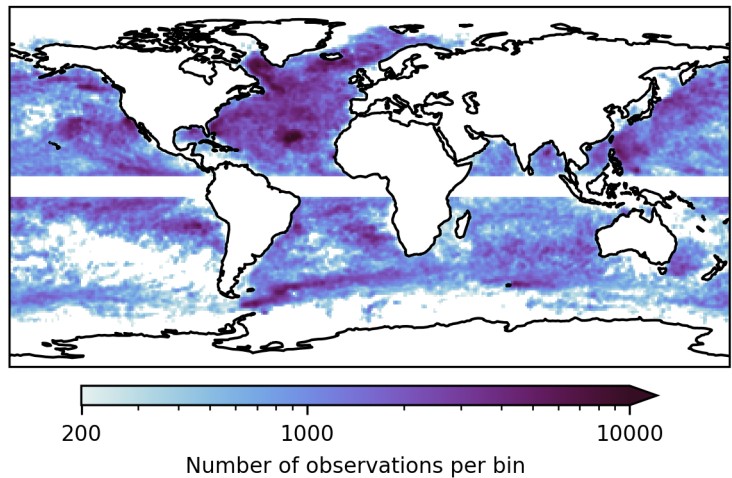

**Figure B1.** Number of drifter observations used for the methods' evaluation per 1° latitude × 1° longitude bin.

*Author contributions.* VB developed the Python package `jaxparrow`, designed and ran the experiments and analysis, and wrote the manuscript. EC proposed the minimization-based cyclogeostrophic inversion formulation. VZDA carried out the initial work on the implementation in `JAX` of the minimization-based cyclogeostrophic inversion. JLS and EC contributed to the design of the experiments, and the writing of the manuscript. JLS, AS and EC contributed to the analysis. JLS and EC acquired funding. All authors reviewed the manuscript.

*Competing interests.* No competing interests are present.

*Acknowledgements.* This research was funded by the French National Space Agency (CNES) through the SWOT Science Team program (SWOT-MIDAS project); by the European Union's Horizon Europe research and innovation programme under the grant No 101093293 (EDITO-Model Lab project).

All the computations presented in this paper were performed using the GRICAD infrastructure (https://gricad.univ-grenoble-alpes.fr), which is supported by Grenoble research communities.

The authors thank Aurélie Albert for processing and making available the eNATL60 surface fields.

The authors would like to thank Maxime Ballarotta, Sammy Metref, and Clément Ubelmann for their feedback on the draft version of this paper.

The authors are also deeply grateful to the two anonymous referees for their thorough reviews. Their constructive comments and suggestions allowed to greatly improved the clarity and quality of the manuscript.

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
