# Peer review of "A Robust Minimization-Based Framework for Cyclogeostrophic Ocean Surface Current Retrieval"

_EGUsphere, 2025_

## Referee Comment (RC2)

This manuscript introduces a new numerical method to solve the cyclogeostrophic balance equation, which is useful for estimating velocity from SSH beyond geostrophic balance. The numerical method phrases the PDE problem as a minimization of the residual and solves it using gradient-based optimization methods, leveraging the capability of JAX. The method is shown to solve the cyclogeostrophic balance equation robustly with minimal noisy artifacts, and the result matches physical expectations. The authors also compared with drifter velocities and showed that the cyclogeostrophic velocity reduces bias for the most energetic regions of the ocean. However, they also note that the resolution of the SSH product also influences the results.

I have used the authors' package jaxparrow, which has existed for a year now on GitHub. From my experience with the package, I agree with the authors' claim that it is a superior way to solve the cyclogeostrophic balance equation. The implementation using JAX is high quality (though I am not an expert in open-source software) and should be the new standard for applying the cyclogeostrophic balance diagnostic of velocity from SSH. Cyclogeostrophic balance has received a revival of interest due to the capability of high-res SSH observation of SWOT [Archer et al., 2025, Tranchant et al., 2025, Tchonang et al., 2025] (additionally, Zhang and Callies [2025, (7)] is essentially the first step of (3)). Unfortunately, jaxparrow was not used in any of these papers. With this paper, jaxparrow will be more widely known to the community. This will improve standardization between implementations of cyclogeostrophic balance and promote cross-validation of results. For this, I recommend the paper for publication. I also recommend that the authors submit another paper to JOSS, where they have a culture of working with the authors to improve the code quality of an open-source package.

With the praise in mind, here are some suggestions that I hope will improve the presentation of the paper.

- 1. The authors name their method "variational", based on calling (4) the "variational form" of the cyclogeostrophic balance equation. I would recommend changing this.
  - "Variational" is a loaded mathematical word. For a nonlinear PDE like (2), the variational formulation does not straightforwardly lead to the minimization of residuals (4). Therefore, naming the method "variational" is not very informative of the underlying principle and can be confusing.
  - I would recommend a different name, say "least-square" or "minimization/optimization based".
- 2. I would also recommend that the author call the traditional method written in (3) "fixed-point". The minimization procedure is also iterative. Therefore, the name "iterative" can be applied to both methods, and can thus be confusing.

  Additionally, the numerical divergence behavior of the traditional method can be understood
  - Additionally, the numerical divergence behavior of the traditional method can be understood through classic theory about the behavior of the fixed-point method and contraction mapping.
- 3. Before (1), I think a quick derivation of the cyclogeostrophic balance equation is helpful.
  - It would show that (1) is based on ignoring the time partial in the horizontal momentum equation.
  - This is not a very well-grounded approximation, especially in the submesoscale, where the time partial term can be equal in magnitude to the advective and the Coriolis terms.
  - I view this as not an issue when the focus is on solving the cyclogeostrophic balance equation. But this is important when comparing with drifter data. Cyclogeostrophic

balance is not obviously better than geostrophic balance in the submesoscale, since it includes one but ignores another term of equal asymptotic order.

- 4. §2.3: I cannot find a description of the minimization procedure.
  - I do note that the authors state more exploration can be done (Line 271), but I want to know what is currently applied.
  - From looking at the code, the authors used SGD in Optax, which I am not familiar with the details. What is the difference from GD in this case, since there is no minibatching?
  - Some math formula that writes out the gradient descent iteration might be helpful when comparing to (3). It is also an opportunity to elucidate how JAX automates the evaluation of the gradient (adjoint) needed in this iteration. This is helpful for an oceanographic audience that is not familiar with these techniques.
- 5. From my point of view, there are two goals this paper tries to achieve. They are both referred to as "accurate" velocity, which is confusing.
  - (a) Showcase a new numerical method for solving the cyclogeostrophic balance equation. It is better than the traditional method
  - (b) Study if cyclogeostrophic balance improves upon geostrophic balance for velocity inversion from SSH.

Then I think the paper has shown (a) convincingly. And I think this should be the main point of the paper, as also stated in the paragraph starting at Line 57.

- One recommendation is to show the SWOT figures in Appendix B in the main text to make the point that cyclogeostrophic balance only depends on local SSH (cf. SQG-based method). Therefore, it can be applied to nontrivial but 2D observation geometry like SWOT swath. It will make mentioning of SWOT appear less gratuitous.
- 6. Point (b) is very complex, and I think it is perpendicular to goal (a). I would like the authors to consider removing goal (b) entirely from this paper and addressing it in a separate piece. But I understand this requires major changes and delays. I leave the authors to decide. With that in mind, I have some comments and questions
  - In recent works Tranchant et al. [2025], Tchonang et al. [2025] have shown in the SWOT context that cyclogeostrophic balance's improvement depends on the region and season. This is consistent with the result in Figure 5.

    The authors should take these papers into account when making the statement "These results suggest that cyclogeostrophic corrections will become increasingly relevant as SSH products achieve higher effective resolution" in Line 257.
  - Figure 5 uses 6-hourly interpolated surface current velocity. This might include NIW signal that the cyclogeostrophic balance will not be able to get (cf. comment 3 above). In general, internal waves and tidal signals might contribute to the error statistics. This fact needs to be mentioned, and better, quantified.
  - When answering this question, model output is your friend. I would like to see Figure A1 plotted as the error of the diagnosed velocity.
     Question: What is the "observed" SSH here? Is there a low-pass filter and added noise to match observations?

A bigger task is to do this for a bigger region and get error statistics like in Figure 4 and 5. The authors can decide if they want to take on this task in the revisions.

- 7. §3.2.2. The error metric can be improved. I agree with Referee #1 on having concerns here.
  - A better metric that is commonly used is mean squared error.
  - Then one can talk about the estimation error of the MSE, where the concept of the standard error can be applied. Right now the term "standard error" is applied erroneously. There should be a division by  $\sqrt{N}$  if the author meant the standard deviation of the sample mean.

**Minor comments**

- 1. Title of §2. Resolution is a loaded word. "The solution to" might be better.
- 2. In (2),  $u_g$  is not defined. I would recommend not using  $u_g$  here, instead use the rotated gradient of  $\eta$ .
- 3. The norm in (5) is better called the little  $\ell_2$  norm, which applies to vectors. The big  $L_2$  norm is usually applied to functions. (4) is an  $L_2$  norm.
- 4. Line 100, 149: make sure to state the *numerical* divergence problem. "Divergence" can be confused with velocity divergence.
- 5. Line 102: "the minimum of J yields a smooth velocity field that does not exhibit the unrealistic features". It is too early in the paper to make this statement. The smoothness of the solution is empirical and can only be inferred from looking at the solution fields in Figure 2. It probably also depends on the SSH data's effective resolution. At this point, there is no reason from the setup that  $u_{cg}$  should be smooth. One can achieve this by using some smoothness regularization, but this is not done here.
- 6. Line 196: "geostrophy can be a *coarse* approximation". Coarse is a loaded word. It could mean it works better for course resolution data. Do you mean that? I don't think Figure 1 is enough to show this.

**References**

- M. Archer, J. Wang, P. Klein, G. Dibarboure, and L.-L. Fu. Wide-swath satellite altimetry unveils global submesoscale ocean dynamics. *Nature*, 640(8059):691–696, Apr. 2025. ISSN 1476-4687. doi: 10.1038/s41586-025-08722-8.
- B. Tchonang, J. Wang, A. F. Waterhouse, A. Lucas, C. G. Griffin, M. R. Archer, L. Kachelein, M. Lankhorst, J. Sevadjian, and L.-L. Fu. SWOT Geostrophic Velocity Validation against in-situ measurements in the California Current, Apr. 2025.
- Y.-T. Tranchant, B. Legresy, A. Foppert, B. Pena-Molino, and H. Phillips. SWOT Reveals Fine-Scale Balanced Motions Driving Near-Surface Currents and Dispersion in the Antarctic Circumpolar Current. *Earth and Space Science*, 12(8):e2025EA004248, 2025. ISSN 2333-5084. doi: 10.1029/2025EA004248.
- X. Zhang and J. Callies. Assessing submesoscale sea surface height signals from the SWOT mission, May 2025.

---

## Author Response (AR1)

**Response to the Referees' Comments**

Vadim Bertrand

**Response to Anonymous Referee 1**

We are grateful to the referee for its detailled comments that helped us improved our manuscript, in particular regarding the choice of the metrics used to assess inversion methods' performances and the existence of a cyclogeostrophic solution. The manuscript has been revised to take the referee comments into consideration.

**Major comments**

**1**

**A critical concern stems from the fact that equation (8) defines the misfit epsilon as the square root of the squared misfit of the two components. This is a positive quantity that should have a Rayleigh distribution. However, the subsequent statistics treat it as a Gaussian quantity in order to determine the standard error of the mean in equation (10). This is not automatically true. More attention needs to be given to the metrics used to assess performance.Ò**

It is indeed true that the previous misfit $\varepsilon$ defined in equation (8) may follow a Rayleigh distribution, $Z = \sqrt{X^2 + Y^2}$, although $X$ and $Y$ may not be formally independant and identically distributed. However, the subsequent statistics are empirical quantities derived from a sample population, without any Gaussian assumption on $\varepsilon$. We believe that the confusion may stem from the incorrect use of the term "standard error", as pointed out by referee 2. The appropriate term should have been "(population) standard deviation of the error". We have changed the definition of $\varepsilon$

**2**

**Because the analysis is based on a Rayleigh distribution, the distinction between the mean (epsilon) and the standard deviation (sigma) is not very meaningful. If the mode of the Rayleigh distribution is defined as sigma (using the notation from Wikipedia—apologies this is a different definition of sigma than the standard deviation), then the mean is sigma sqrt(pi/2), and the standard deviation is sigma sqrt((4-pi)/2). This means that Figure 4 and Figure C2 present redundant information. Moreover, because the mean and the standard deviation convey the same information, it's not clear how to interpret the information in Figures 4 and C2.**

We agree with the referee that presenting both the mean and the standard deviation of the misfit can be redundant, as they convey similar information when the underlying distribution is close to Rayleigh. To address this issue, we have revised our approach to focus on a single, clearer performance indicator. We now use the squared misfit and compute the RMSE, which captures both bias and variability in a single, well-known quantity. As a result, Figure 4 has been updated accordingly, and Figure C2 has been removed to avoid redundancy and improve clarity.

**3**

**The manuscript uses surface drifters as a reference for assessing whether the cyclogeostrophic velocity provides a more realistic assessment of total velocity, but the plotted results do not provide compelling evidence. Figure 4a shows that cyclogeostrophic velocities have greater**

spread relative to drifters in regions of high kinetic energy, but this does not show whether the means have converged. Figure C2a shows that there is on average a non-zero difference from drifters and that the mean speed difference is larger in regions of high eddy energy, which seems unsurprising. Figure 4b shows that for NeurOST the spread of the corrected velocities is smaller in eddy-intense regions such as the Kuroshio Extension, while Figure 4c shows that with DUACS data the spread is slightly larger in eddy-intense regions. This does not formally tell us whether the cyclogeostrophic correction brings the altimeter-derived currents into better agreement with drifter velocities. Figures C2b and C2c should contain information showing a reduction in misfit with the cyclogeostrophic correction, but they are not discussed in the text. Figure 5 shows misfit but does not indicate a statistically significant improvement from the cyclogeostrophic correction. The manuscript needs to provide a clear metric showing that the variational method works.

We acknowledge that using separately the population mean and standard deviation of the misfit can be unclear and may obscure the demonstration that the variational method is effective. To clarify the analysis, we now square the misfit in equation (8) and rely solely on the RMSE to assess the performance of the inversion methods, as also suggested by referee 2. This approach simultaneously accounts for both the accuracy and the precision of the reconstruction. As a result, Figure 4 has been updated (along with the corresponding text), and Figure C2 has been removed.

**4**

Equation (11) is based on the fraction of standard deviation explained by the correction. A more standard metric would look at fraction of variance (using a squared quantity). The manuscript should provide more clarity about the choice of statistical metrics and the robustness of the metrics reported in the paper.

In the revised manuscript, we now base our comparison on the squared misfit and report the corresponding RMSE, which is equivalent to using the variance of the residuals as a performance metric. This modification makes the comparison between methods more standard and statistically robust.

**5**

The authors cite a few studies but do not revisit the analytic gradient wind solution, discussed nicely by Penven et al (2014) and by Knox and Ohmann (2006, https://doi.org/10.1016/j.cageo.2005.09.009). To me this seems like a disappointing gap. Of note, the gradient wind approach allows an analytic solution, which is imperfect but presumably could serve as a first guess for the iterative or variational approaches. Readers who want to implement a robust algorithm will probably want to understand the performance of the variational approach in the context of the gradient wind solution and to understand whether the gradient wind solution is useful as an initial guess for other solution strategies.

We agree that the gradient wind formulation discussed by Penven et al. (2014) and Knox and Ohmann (2006) provides an elegant analytical framework that can, in some cases, serve as a useful reference or initial approximation.
However, the gradient wind equation presents two main limitations that motivated our choice not to rely on it in the present study. First, it may have no real solution under certain dynamical conditions, particularly when the curvature or pressure gradient terms lead to non-physical results. Second, its formulation depends on the radius of deformation, which is not known a priori and can vary significantly in space and time. These aspects make the method difficult to apply in a global and automated context, which is why we chose to focus instead on the iterative approach, as in Cao et al. (2023) and Ioannou et al. (2019).
Nonetheless, we agree that the analytical solution is valuable for understanding when the cyclogeostrophic balance admits a physical solution, and that it may serve as a useful initial guess for iterative or variational inversion methods. We have therefore added a dedicated section presenting the gradient-wind solution and discussing its limitations, as well as a note in the discussion section highlighting its potential use as a first-guess strategy for future work.

**6**

**Lines 40 and 44. "convective force". By convention, in oceanography convection is buoyancy driven, and horizontal motions are advective. They are not strictly a force, since the advective terms are intrinsic to the Navier Stokes equations and not imposed externally. This usage in oceanography contrasts with some fluid mechanics literature, which can distinguish between convection (i.e. advection) vs natural convection (motion driven by buoyancy gradients). To ensure that the manuscript is accessible to readers with an oceanographic background, the term "convective force" should be replaced with "advective terms".**

We agree that the term "convective force" is not appropriate in an oceanographic context, where horizontal motions are typically referred to as advective terms rather than as a force. We have therefore replaced "convective force" with "advective terms" throughout the manuscript to ensure consistency with standard oceanographic usage.

**7**

**Line 44. "Coriolis force". Coriolis is usually described as a "pseudo force". It would be more accurate to say "Coriolis term".**

Similarly, we acknowledge that the Coriolis acceleration is more accurately described as a pseudo force and should be referred to as a "Coriolis term." The manuscript has been revised accordingly.

**8**

**Lines 80-81. "A typical case of divergence is when the cyclogeostrophic equation has no solution". It would be useful to clarify why the equation sometimes has no solution.**

We agree that this statement required further clarification. The cyclogeostrophic equation may have no real solution when the curvature and pressure gradient terms lead to an imbalance such that the discriminant of the quadratic form becomes negative. This typically occurs in regions where the centrifugal acceleration exceeds the pressure gradient force, preventing the existence of a dynamically consistent steady-state velocity. We have added a section mentionning the analytical gradient wind solution in which we clarify this point.

**9**

**Line 140. "to drogued SVP-type drifters". Drifters are notorious for losing their drogues without being properly flagged as missing drogues. It would be helpful to remind readers which version of the data you are using and to specify how well you think the drogue losses are flagged.**

In the Global Drifter Program (GDP) database, an additional flag is provided to indicate the presence of the drogue. This flag is determined following the procedure described by Lumpkin et al. (2013), in which the data are automatically reanalyzed to detect drogue loss based on anomalous downwind ageostrophic motion. In our analysis, we retain only the observations identified as drogued according to this procedure. We have clarified this point in the revised manuscript.

**10**

**Line 145. "Due to the use of Arakawa C-grids". How is the C-grid being used? Gridded data products are not intrinsically on the C-grid, so this point requires clarification.**

As explained in Section 2.3 of the manuscript, partial derivatives are computed using finite differences on an Arakawa C-grid. In this configuration, the sea surface height (SSH) is defined at the scalar (T) points of the grid, while the zonal and meridional velocity components ($u$ and $v$) are defined at the staggered C-grid points. Consequently, when computing derived quantities such as velocity magnitude or vorticity, an interpolation step is required to ensure that all variables are collocated on the same grid points. Specifically, for the velocity magnitude, $u$ and $v$ are first interpolated to the T points prior to computation, whereas for vorticity, the

calculation is performed on the C-grid and the resulting field is then interpolated back to the T points. We have clarified this explanation in the revised manuscript to make the use of the C-grid more explicit.

**11**

**Equation (8). In calculating epsilon, what is the spatial separation allowed for the interpolation? Are there constraints on satellite overpass time or distance from ground track?**

In our analysis, we do not apply any constraint on the temporal or spatial separation between drifter observations and satellite overpasses, since our objective is to compare the inversion methods using Level-4 (L4) gridded products rather than along-track data. Consequently, the interpolation of the model velocity field $\mathbf{u}_M$ at the drifter position $\mathbf{X}_i$ is performed directly on the regular L4 grid, and the effective spatial separation is therefore bounded by $\frac{\sqrt{dx^2+dy^2}}{2}$, with $dx$ and $dy$ denoting the grid spacings in the zonal and meridional directions.

**12**

**Equations (10-11). Is N the same for M1 and M2? If it different, the normalized difference in equation (11) could depend largely on the number of samples, which is not really the intent of this metric, I believe. On the other hand, if N is the same in both cases, then this is really a comparison of rms error in the two cases. That leads to a question of whether the variance would be more appropriate as a metric.**

Within each spatial–temporal bin, the binning domain and the set of drifter observations are identical for all methods and products, even though the gridded datasets may have different spatial resolutions. In other words, all comparisons are performed within the same geographical and temporal bins, ensuring that $N$ is identical for $M_1$ and $M_2$. Consequently, the normalized difference in equation (10) does not depend on the number of samples; it reflects only differences in the dispersion of the misfit between methods.
In line with the referee's suggestion, we have also revised the analysis to work with the squared misfit and to report RMSE (and, equivalently, percent reduction in MSE) as our primary metric.

**13**

**Line 213. "anticyclonic (cyclonic)". Is this redundant with the previous sentence? Classic warm core rings are anticyclonic and are north of the Gulf Stream. Cold core rings are cyclonic and are south of the*** Gulf Stream.\*\***

We agree that the use of "anticyclonic (cyclonic)" could appear redundant given the description of the Gulf Stream context. However, our intent was to distinguish between meanders of the Gulf Stream jet and detached eddies, which are distinct features. To avoid confusion, we have clarified the text to refer to the northern (southern) meanders of the Gulf Stream rather than branches, and we now explicitly state that similar differences are observed for anticyclonic (cyclonic) eddies. This distinction makes it clear that the statement refers to both the jet meanders and the surrounding eddy field.

**14**

**Line 216. "introduces artifacts in the most dynamic parts of the jet and eddies." More explanation would help. What accounts for the artifacts? Does the variational method implicitly impose a smoothness parameter by minimizing the global misfit?**

The artifacts observed in the most dynamic regions of the jet and eddies arise primarily from the local convergence behavior of the fixed-point iterative scheme, see also referee 2 comment (2). In these regions, strong curvature and nonlinearity can cause the algorithm to converge toward local, non-physical solutions or to fail to converge entirely if the initial guess does not lie within the basin of attraction of a stable solution. This results in locally inconsistent velocity estimates that appear as small-scale artifacts.
Regarding the second point, the variational formulation does implicitly impose a degree of spatial smoothness,

since it minimizes a global misfit functional (involving the computation of spatial derivatives) that integrates over the entire domain. While no explicit regularization term is introduced, the minimization process itself tends to favor spatially coherent solutions. We have expanded the discussion in the revised manuscript to clarify these two aspects.

**15**

**Line 237. "NeurOST-based cyclogeostrophy clearly reduces standard error". Given the size of the standard errors plotted in Fig. 5, does a reduction in standard error show that the results are better or just that they are slightly more consistent, although not at a level that could be judged to be statistically significant?**

We agree that the original figure was difficult to interpret and that the reduction in standard error, as previously presented, did not clearly demonstrate a statistically significant improvement of the cyclogeostrophic reconstruction compared to the geostrophic one.
To address this, we have revised the analysis by applying a logistic regression model that estimates the conditional probability $P(\varepsilon_{cg} < \varepsilon_g \mid \|u_c\|)$, representing the probability that the cyclogeostrophic reconstruction outperforms the geostrophic one as a function of the cyclostrophic correction magnitude.

**16**

**Lines 240-241. "we find that at the highest EKE percentiles, cyclogeostrophy reduces reconstruction uncertainty by nearly 10 % upon geostrophy when employing the variational method." Changes shown in Figure 5 do not appear statistically significant and would not pass a Student T-test. This does not appear to be a robust measure of the success of the algorithm, or even of the relevance of the cyclogeostrophic correction. That's not to say that it's not useful, but the presentation will need to be reworked.**

We agree that the differences shown in the original version of Figure 5 were not statistically significant and could not be considered a reliable indicator of the success of the variational method. To address this, we have replaced the standard error comparison with a logistic regression which provides a direct and testable measure of improvement. This approach enables the derivation of confidence bands, clearly identifying where the cyclogeostrophic correction statistically improves the reconstruction, particularly in regions of strong cyclostrophic activity.

**17**

**Figure 5. Reiterating my previous point, the results in the figure suggest no statistically significant improvement by using cyclogeostrophy compared with geostrophy, and no visible differences except in high energy areas, and then only for the NeurOST product. As discussed in the text, DUACS doesn't benefit from correction and is not as good as NeurOST. However, the standard error is so large that differences are not formally detectable, even after screening for high energy areas**

We agree that the previous version of Figure 5 did not allow a statistically meaningful interpretation of the improvement brought by the cyclogeostrophic correction. In the revised version, Figure 5 now presents the probability obtained from the logistic regression, together with its confidence intervals, offering a clearer and statistically more rigorous view of where and to what extent the cyclogeostrophic reconstruction outperforms the geostrophic one, especially for the NeurOST product in dynamically active regions.

**18**

**Line 286. "can be readily integrated as a modular component into such DA systems." It's not clear why this would be needed in an adjoint-based data assimilation system, since the model and adjoint should account for the distinctions between geostrophic and total velocity terms.**

We agree that, in a classical adjoint-based data assimilation (DA) framework, the distinction between geostrophic and total velocity components is already embedded within the model's dynamical core and its adjoint, so a separate cyclogeostrophic module would not be strictly necessary.

Our intent was rather to emphasize that the proposed formulation, being fully differentiable and implemented in JAX, can be flexibly integrated as an additional or alternative constraint within variational or machine-learning–based DA systems. In particular, its modular structure allows the introduction of custom penalization terms, for example to jointly filter noisy SSH observations (such as those from SWOT, see Tranchant et al. (2025)) or to impose additional similarity constraints with respect to ancilary data such as sea surface temperature (see Le Guillou, Chapron, and Rio (2025)).

We have clarified this point in the revised text to make it clear that our approach is not meant to replace the dynamics already captured by adjoint-based systems, but rather to offer a lightweight, differentiable component that can be incorporated into emerging DA or learning frameworks to improve the handling of nonlinear balance relations and observation noise.

**19**

**Appendix A. "We choose snapshots of the Alboran Sea as it features two large and persistent gyres subject to cyclogeostrophy." In Figure A1, there are clear distinctions between the iterative and variational approaches, but are there skill metrics to quantify these differences? What specific aspects of this account for the distinctions between the variational and iterative approaches?**

Following the suggestions of referee 2, we have extended the evaluation by including in the main text a quantitative skill assessment using the high-resolution eNATL60 simulation as a reference and pseudo-SWOT observation data, following a similar methodology as the updated drifter-based comparison.

Furthermore, we have expanded the discussion to better explain the origins of these distinctions. As clarified in response to comment 14 and to referee 2's comments 2, 3, and minor comment 5, the main differences arise from the convergence behavior of the iterative fixed-point solver and the global nature of the variational minimization, which implicitly imposes smoother and dynamically more coherent corrections. These points are now explicitly discussed in the revised manuscript and illustrated in the updated figures.

**20**

**Appendix B. The SWOT inversion is interesting, but there isn't much context, nor is there a set of in situ measurements to use to assess the skill. What science results emerge the from the SWOT demonstration? Is the SWOT discussion needed in this manuscript?**

We acknowledge that, in the previous version, the SWOT demonstration does not include a direct validation against in situ measurements and therefore provides limited quantitative assessment of skill. Our intent was primarily to illustrate the applicability of the proposed inversion framework to SWOT-like swath SSH data (L3 products), where the departure from geostrophy is expected to be significant.

As also noted by referee 2, there is a growing interest within the SWOT community in moving beyond the geostrophic approximation by directly exploiting the high-resolution SSH fields from the swath (Archer et al. 2025; Tranchant et al. 2025; Wang et al. 2025; Zhang and Callies 2025). In this context, we believe that showing the feasibility of the cyclogeostrophic inversion on SWOT data is relevant.

Following referee 2's comments 5 and 6, we have replaced the SWOT appendix by a quantitative assesment of the cyclogeostrophic reconstruction methods based on the eNATL60 model and pseudo-SWOT swaths. In the revised manuscript, we reconstruct cyclogeostrophic currents from pseudo-SWOT swaths over the Mediterranean Sea and compare them with the model's true surface field.

**21**

**Appendix C. As noted above, the mean misfit in Appendix C presents some fundamental challenges. In addition, the wording in the figure caption is unclear. What is meant by spatial binning? I think it would be sufficient to say "but showing the mean misfit instead of the standard error of the misfit". Ideally all four panels should be discussed, although as noted**

above, the interpretation is not clear. The mean misfit in panel (a) is fairly large and by definition always positive. In panel (b) the cyclogeostrophic approach shows a decrease in speed bias in western boundary currents and an increase in speed bias near the equator. In panel (c) DUACS shows an increase in speed bias in western boundary currents. Panel (d) shows better results with NeurOST relative to DUACS geostrophy, but that's presumably mostly a reflection of differences between NeurOST and DUACS.

We thank the referee for this detailed comment. As suggested, we have removed Appendix C in the revised version of the manuscript, since we now rely exclusively on the RMSE as the performance metric. This modification addresses the interpretational issues associated with the mean misfit and simplifies the presentation of the results, making the assessment more consistent and statistically robust.

**22.**

I found the package name, jaxparrow, to be a massive distraction. As the authors are probably aware, Jack Sparrow is the name of the pirate protagonist in the Disney film, Pirates of the Caribbean. The name Jack Sparrow is also associated with the 16th to 17th century pirate Jack Ward. There has been quite a bit of historical scholarship on pirates in recent years. I'm not sure if the authors if this paper intended to invoke the Disney film or the historical antecedents. Regardless, the name was a distraction for me, and it left me asking a broad range of questions that have nothing to do with the content of the manuscript. Does the name of the software package glorify a Disney film at a moment in time when people have been asking if they should boycott Disney? Would Disney protest usage of the name, citing concerns about trademark or copyright? (I don't think they should, given the spelling change and given the historical origins of the name that long pre-date the Disney corporation, but Disney has been a notoriously fierce defender of its trademarks.) Does the name glorify pirates, who in the modern world have been antagonists to ocean observation? All of these questions, I leave to others (and the lawyers) to untangle. I merely remind the authors that their cute choice of name comes at the cost of pulling reader attention away from the core concepts.

We thank the referee for sharing this perspective. We acknowledge that the package name jaxparrow may inadvertently distract some readers or evoke unintended associations. The name has been initally derived from a contraction of JAX–the differentiable programming framework used for implementation–and arrow, referring to the velocity vector central to the method. No references to Disney or any historical pirate figure was intended. It became `jaxparrow` as a humorous nod to a popular character.

**Minor comments**

We thank the referee for these detailed and helpful editorial suggestions. We agree with all the proposed corrections except for point 1. According to the SI and Ocean Science submission guidelines (i.e. "Spaces must be included between number and unit (e.g. 1 %, 1 m)."), the percent sign (%) is treated as a unit and should therefore be preceded by a space (e.g., 20 % rather than 20%). All other points have been addressed as suggested in the revised manuscript.

**Response to Anonymous Referee 2**

We thank the referee for the constructive and encouraging assessment of our work, as well as for the insightful suggestions that helped us clarify and improve the manuscript, especially with the addition of an evaluation using the eNATL60 model and pseudo-SWOT observations.
We particularly appreciate the referee's positive feedback on the implementation quality of jaxparrow and its potential contribution to the standardization of cyclogeostrophic balance diagnostics. Regarding the suggestion to submit to JOSS, we note that we have already considered this option, but the package was deemed out of scope because its codebase is too small (not enough lines of code) to meet JOSS publication criteria.

**Major comments**

**1**

**The authors name their method "variational", based on calling (4) the "variational form" of the cyclogeostrophic balance equation. I would recommend changing this.**

- **"Variational" is a loaded mathematical word. For a nonlinear PDE like (2), the variational formulation does not straightforwardly lead to the minimization of residuals (4). Therefore, naming the method "variational" is not very informative of the underlying principle and can be confusing.**
- **I would recommend a different name, say "least-square" or "minimization/optimization based".**

It is indeed true that the term "variational" can be misleading, as it carries a specific mathematical meaning that may not strictly apply to our formulation. Although our approach allows for the inclusion of arbitrary regularization terms—which would make it closer in spirit to a true variational formulation—we agree that "minimization-based" better conveys the nature of our method. We have therefore replaced the term variational with minimization-based throughout the revised manuscript.

**2**

**I would also recommend that the author call the traditional method written in (3) "fixed-point". The minimization procedure is also iterative. Therefore, the name "iterative" can be applied to both methods, and can thus be confusing. Additionally, the numerical divergence behavior of the traditional method can be understood through classic theory about the behavior of the fixed-point method and contraction mapping.**

We agree with the referee that calling the traditional method written in equation (3) "iterative" is ambiguous, since both approaches involve iterative procedures. The suggestion to refer to this approach as a fixed-point method is well taken. This terminology is more precise and aligns with the mathematical framework of contraction mappings that explains its possible divergence. We have adopted this terminology in the revised manuscript.

Before (1), I think a quick derivation of the cyclogeostrophic balance equation is helpful.

- It would show that (1) is based on ignoring the time partial in the horizontal momentum equation.
- This is not a very well-grounded approximation, especially in the submesoscale, where the time partial term can be equal in magnitude to the advective and the Coriolis terms.
- I view this as not an issue when the focus is on solving the cyclogeostrophic balance equation. But this is important when comparing with drifter data. Cyclogeostrophic balance is not obviously better than geostrophic balance in the submesoscale, since it includes one but ignores another term of equal asymptotic order.

We acknowledge that including a brief derivation of the cyclogeostrophic balance equation before equation (1) improves clarity. We have added this derivation, explicitly showing that it results from neglecting the local time-derivative term in the horizontal momentum equations. We also note that this approximation may not hold at submesoscales, where the time-derivative term can be comparable to the advective and Coriolis terms. This limitation is now explicitly discussed in relation to the drifter-based validation.

**§2.3: I cannot find a description of the minimization procedure.**

- I do note that the authors state more exploration can be done (Line 271), but I want to know what is currently applied.
- From looking at the code, the authors used SGD in Optax, which I am not familiar with the details. What is the difference from GD in this case, since there is no minibatching?
- Some math formula that writes out the gradient descent iteration might be helpful when comparing to (3). It is also an opportunity to elucidate how JAX automates the evaluation of the gradient (adjoint) needed in this iteration. This is helpful for an oceanographic audience that is not familiar with these techniques.

We agree with the referee that a clearer description of the minimization procedure is necessary. We include an explicit formula for the gradient descent update and clarify how JAX is used to automatically compute the required gradients. This addition makes the procedure more transparent and helps readers unfamiliar with automatic differentiation understand how the optimization is implemented. The optimization is indeed performed by default using the stochastic gradient descent (SGD) implementation from Optax, without mini-batching, which effectively reduces to a standard gradient descent iteration.

From my point of view, there are two goals this paper tries to achieve. They are both referred to as "accurate" velocity, which is confusing.

(a) Showcase a new numerical method for solving the cyclogeostrophic balance equation. It is better than the traditional method

(b) Study if cyclogeostrophic balance improves upon geostrophic balance for velocity inversion from SSH.

Then I think the paper has shown (a) convincingly. And I think this should be the main point of the paper, as also stated in the paragraph starting at Line 57.

- One recommendation is to show the SWOT figures in Appendix B in the main text to make the point that cyclogeostrophic balance only depends on local SSH (cf. SQG-based method). Therefore, it can be applied to nontrivial but 2D observation geometry like SWOT swath. It will make mentioning of SWOT appear less gratuitous.

Regarding the suggestion to move the SWOT results from Appendix B into the main text, we prefer not to include the existing appendix figures directly, since their purpose is illustrative rather than evaluative.

Instead, motivated by this comment and by comment 6, we now include in the main text a new assessment based on the eNATL60 model and pseudo-SWOT observations in the swath. This addition demonstrates the applicability of our approach to the SWOT observation geometry while providing a more rigorous comparison than the original appendix material. The SWOT appendix as therefore been removed.

**6**

**Point (b) is very complex, and I think it is perpendicular to goal (a). I would like the authors to consider removing goal (b) entirely from this paper and addressing it in a separate piece. But I understand this requires major changes and delays. I leave the authors to decide. With that in mind, I have some comments and questions**

- **In recent works Tranchant et al. [2025], Tchonang et al. [2025] have shown in the SWOT context that cyclogeostrophic balance's improvement depends on the region and season. This is consistent with the result in Figure 5. The authors should take these papers into account when making the statement "These results suggest that cyclogeostrophic corrections will become increasingly relevant as SSH products achieve higher effective resolution" in Line 257.**
- **Figure 5 uses 6-hourly interpolated surface current velocity. This might include NIW signal that the cyclogeostrophic balance will not be able to get (cf. comment 3 above). In general, internal waves and tidal signals might contribute to the error statistics. This fact needs to be mentioned, and better, quantified.**
- **When answering this question, model output is your friend. I would like to see Figure A1 plotted as the error of the diagnosed velocity. Question: What is the "observed" SSH here? Is there a low-pass filter and added noise to match observations? A bigger task is to do this for a bigger region and get error statistics like in Figure 4 and 5. The authors can decide if they want to take on this task in the revisions.**

We agree that fully removing the drifter-based evaluation would require a major restructuring of the manuscript, and we have therefore opted to retain this evaluation. However, we have strengthened and clarified this part of the analysis in several ways.

First, we now explicitly reference Tranchant et al. (2025) in line 257, noting that the improvement from cyclogeostrophic corrections depends on the effective SSH resolution, consistent with our findings. Tchonang et al. (2025) is refered to earlier in the manuscrit to underline the interest of the community in reconstructing cyclogeostrophic currents in the SWOT swath.

Second, we now apply the same pre-processing to the drifters data as Müller et al. (2019) by (i) removing the Ekman contribution (obtained from GlobCurrent, implementing Rio, Mulet, and Picot (2014)) and (ii) filtering NIW signal using a second order Butterworth filter.

Finally, we have added a complementary evaluation using the eNATL60 simulation. In the revised manuscript, we reconstruct geostrophic and cyclogeostrophic currents from pseudo-SWOT swaths over the Mediterranean Sea and compare them with the model's true surface field. This additional analysis reinforces point (b). In this experiment, we do not apply a low-pass filter or add noise to the SSH field: the eNATL60 SSH is directly reinterpolated onto a SWOT-like swath at 2 km resolution.

**7**

**§3.2.2. The error metric can be improved. I agree with Referee #1 on having concerns here.**

- **A better metric that is commonly used is mean squared error.**
- **Then one can talk about the estimation error of the MSE, where the concept of the standard error can be applied. Right now the term "standard error" is applied erroneously. There should be a division by $\sqrt{N}$ if the author meant the standard deviation of the sample mean.**

We agree with the referee and with Referee #1 that the definition of the error metric needed improvement. The analysis now uses the root mean square error (RMSE) as the main performance metric. This change

ensures that the assessment is statistically sound and avoids the incorrect use of the term "standard error".

**Minor comments**

We agree with all the minor comments, which have been implemented in the revised manuscript.

**References**

Archer, Matthew, Jinbo Wang, Patrice Klein, Gerald Dibarboure, and Lee-Lueng Fu. 2025. "Wide-Swath Satellite Altimetry Unveils Global Submesoscale Ocean Dynamics." *Nature* 640 (8059): 691–96. https://doi.org/10.1038/s41586-025-08722-8.

Cao, Yuhan, Changming Dong, Alexandre Stegner, Brandon J. Bethel, Chunyan Li, Jihai Dong, Haibin Lü, and Jingsong Yang. 2023. "Global Sea Surface Cyclogeostrophic Currents Derived From Satellite Altimetry Data." *Journal of Geophysical Research: Oceans* 128 (1): e2022JC019357. https://doi.org/10.1029/2022JC019357.

Ioannou, Artemis, Alexandre Stegner, Alexandre Tuel, Briac LeVu, Franck Dumas, and Sabrina Speich. 2019. "Cyclostrophic Corrections of AVISO/DUACS Surface Velocities and Its Application to Mesoscale Eddies in the Mediterranean Sea." *Journal of Geophysical Research: Oceans* 124 (12): 8913–32. https://doi.org/10.1029/2019JC015031.

Knox, John A., and Paul R. Ohmann. 2006. "Iterative Solutions of the Gradient Wind Equation." *Computers & Geosciences* 32 (5): 656–62. https://doi.org/10.1016/j.cageo.2005.09.009.

Le Guillou, Florian, Bertrand Chapron, and Marie-Helene Rio. 2025. "VarDyn: Dynamical Joint-Reconstructions of Sea Surface Height and Temperature From Multi-Sensor Satellite Observations." *Journal of Advances in Modeling Earth Systems* 17 (4): e2024MS004689. https://doi.org/10.1029/2024MS004689.

Lumpkin, Rick, Semyon A. Grodsky, Luca Centurioni, Marie-Helene Rio, James A. Carton, and Dongkyu Lee. 2013. "Removing Spurious Low-Frequency Variability in Drifter Velocities." *Journal of Atmospheric and Oceanic Technology* 30 (2): 353–60. https://doi.org/10.1175/JTECH-D-12-00139.1.

Müller, Felix L., Denise Dettmering, Claudia Wekerle, Christian Schwatke, Marcello Passaro, Wolfgang Bosch, and Florian Seitz. 2019. "Geostrophic Currents in the Northern Nordic Seas from a Combination of Multi-Mission Satellite Altimetry and Ocean Modeling." *Earth System Science Data* 11 (4): 1765–81. https://doi.org/10.5194/essd-11-1765-2019.

Penven, Pierrick, Issufo Halo, Stéphane Pous, and Louis Marié. 2014. "Cyclogeostrophic Balance in the Mozambique Channel." *Journal of Geophysical Research: Oceans* 119 (2): 1054–67. https://doi.org/10.1002/2013JC009528.

Rio, M.-H., S. Mulet, and N. Picot. 2014. "Beyond GOCE for the Ocean Circulation Estimate: Synergetic Use of Altimetry, Gravimetry, and in Situ Data Provides New Insight into Geostrophic and Ekman Currents." *Geophysical Research Letters* 41 (24): 8918–25. https://doi.org/10.1002/2014GL061773.

Tchonang, Babette, Jinbo Wang, Amy Frances Waterhouse, Andrew Lucas, Caeli Griffin Griffin, Matthew Robert Archer, Luke Kachelein, Matthias Lankhorst, Jeffrey Sevadjian, and Lee-Lueng Fu. 2025. "SWOT Geostrophic Velocity Validation Against in-Situ Measurements in the California Current." *ESS Open Archive*, April. https://doi.org/10.22541/essoar.174554354.40247813/v1.

Tranchant, Y.-T., B. Legresy, A. Foppert, B. Pena-Molino, and H. Phillips. 2025. "SWOT Reveals Fine-Scale Balanced Motions Driving Near-Surface Currents and Dispersion in the Antarctic Circumpolar Current." *Earth and Space Science* 12 (8): e2025EA004248. https://doi.org/10.1029/2025EA004248.

Wang, Jinbo, Andrew J. Lucas, Scott Stalin, Matthias Lankhorst, Uwe Send, Oscar Schofield, Luke Kachelein, et al. 2025. "SWOT Mission Validation of Sea Surface Height Measurements at Sub-100 Km Scales." *Geophysical Research Letters* 52 (11): e2025GL114936. https://doi.org/10.1029/2025GL114936.

Zhang, Xihan, and Jörn Callies. 2025. "Assessing Submesoscale Sea Surface Height Signals From the SWOT Mission." *Journal of Geophysical Research: Oceans* 130 (10): e2025JC022879. https://doi.org/10.1029/2025JC022879.

---

## Referee Report (RR1)

The newer version of this manuscript, in my opinion, flows much better. The scientific content is also much improved. I recommend its publication.

Here are some minor suggestions:

1. Line 37: The paper of Dù et al. [2025] might be worth mentioning here. It is a modeling work that extends the geostrophic balance. Although it does not directly involve SWOT data, it investigates the theory behind the common approximations, including cyclogeostrophc balance. For example, it elucidates the connection between Zhang and Callies [2025] and the cyclogeostrophc balance.

2. Line 45: the term "ageostrophic" and "not capture by the *balance*" is confusing.

3. Line 137: This sentence is confusing. Do you mean it can be written as an $L^2$ norm, without the integral? I do not see how it is used in later "further discussions".

4. Line 236: Here, a discussion of the error of SWOT SSH is warranted. For the purpose of showcasing the method, it is ok to not add noise into the simulated data. But SWOT observation is definitely not going to be the same as the model data. Error or SWOT is discussed in many papers, including Wang et al. [2019], Peral et al. [2024], Zhang and Callies [2025] for a non-exhaustive list.

5. (14): for the model case, the sum should also include over the domain points. So $N$ should be much larger than 31.

6. Line 342: In fact Tchonang et al. [2025]'s results show cyclogeostrophy did not improve over geostrophic balance. One should be careful when citing.

7. Line 355: Standard error made a sneaky comeback. This is perhaps a typo.

8. Figure 4: Is it correct to call these RMSE? What mean?

**References**

R. S. Dù, K. S. Smith, and O. Bühler. Next-Order Balanced Model Captures Submesoscale Physics and Statistics. *Journal of Physical Oceanography*, 55(10):1679–1697, Sept. 2025. ISSN 0022-3670, 1520-0485. doi: 10.1175/JPO-D-24-0146.1.

E. Peral, D. Esteban-Fernández, E. Rodríguez, D. McWatters, J.-W. De Bleser, R. Ahmed, A. C. Chen, E. Slimko, R. Somawardhana, K. Knarr, M. Johnson, S. Jaruwatanadilok, S. Chan, X. Wu, D. Clark, K. Peters, C. W. Chen, P. Mao, B. Khayatian, J. Chen, R. E. Hodges, D. Boussalis, B. Stiles, and K. Srinivasan. KaRIn, the Ka-Band Radar Interferometer of the SWOT Mission: Design and in-Flight Performance. *IEEE Transactions on Geoscience and Remote Sensing*, 62: 1–27, 2024. ISSN 1558-0644. doi: 10.1109/TGRS.2024.3405343.

B. Tchonang, J. Wang, A. F. Waterhouse, A. Lucas, C. G. Griffin, M. R. Archer, L. Kachelein, M. Lankhorst, J. Sevadjian, and L.-L. Fu. SWOT Geostrophic Velocity Validation against in-situ measurements in the California Current, Apr. 2025.

J. Wang, L.-L. Fu, H. S. Torres, S. Chen, B. Qiu, and D. Menemenlis. On the Spatial Scales to be Resolved by the Surface Water and Ocean Topography Ka-Band Radar Interferometer. *Journal of Atmospheric and Oceanic Technology*, 36(1):87–99, Jan. 2019. ISSN 0739-0572, 1520-0426. doi: 10.1175/JTECH-D-18-0119.1.

X. Zhang and J. Callies. Assessing Submesoscale Sea Surface Height Signals From the SWOT Mission. *Journal of Geophysical Research: Oceans*, 130(10):e2025JC022879, 2025. ISSN 2169-9291. doi: 10.1029/2025JC022879.

---

## Author Response (AR2)

**Response to the Referees' Comments**

Vadim Bertrand

**Response to Anonymous Referee 1**

We thank the reviewer for their careful follow-up and for the helpful minor suggestions, which we have addressed to further improve the manuscript.

**1**

**Line 192. The DUACS spatio-temporal grid does not represent the formal resolution of the data. Maybe the text should say, "DUACS provides data at daily temporal increments on a 1/8 degree spatial grid."**

The revised manuscript has been modified accordingly.

**2**

**Figure 1. "Maps of deviation from cyclogeostrophy". I understand NeurOST to be an ocean surface topography product. What constitutes the truth for this product in order to determine deviation from cyclogeostrophy? Is the minimization-based method assumed to be the truth?**

There is no "truth" here, we simply show the evaluation of the cyclogeostrophic imbalance from Eq. 9. In the revised manuscript, the caption now state more clearly: "Maps of cyclogeostrophic imbalance, computed from Eq. 9 …"

**3**

**Line 338. "As discussed in Sections 2.2 and 2.3, these differences are likely linked to the distinct mathematical nature of the two approaches." As written this could be interpreted to say that the mathematical nature of the two approaches is the same for both and distinct from other methods? If the intent is to say that there are mathematical distinctions between the two approaches, then it would be good to say that directly: for example, "As discussed in Sections 2.2 and 2.3, the differences are likely linked to the mathematical distinctions between the two approaches."**

The revised manuscript has been modified accordingly.

**4**

**Figure 5. The distinctions between panels b and c are difficult to see. I think the authors want readers to notice that there is more blue in panel b and more red in panel c, but that point is not universally consistent. The text at line 398 points to western boundary current regions, but these regions are small are difficult to see. Would an inset panel help to highlight the relevant differences?**

Indeed. In the revised manuscript, we have added insets to Fig. 5 and refer to them in the text.

**5**

**Line 194. "data has" -> "data have"**
**Line 347. "results suggests" -> "results suggest"**
**Line 370.  The reference to three previous studies, separated by semi-colons, doesn't read cleanly. Maybe a better phrasing would be, "Consistent with previous work (e.g. Archer et al, Tchonang et al, Tranchant et al), ...."**
**Line 401.  Missing word. Change to "rather than geostrophy"**

The revised manuscript has been modified accordingly.

**Response to Anonymous Referee 2**

We thank the reviewer for their positive assessment of the revised manuscript, noting the improved clarity and scientific content. We have carefully considered their minor suggestions and addressed them in the revised version.

**1**

**Line 37: The paper of Dù et al. [2025] might be worth mentioning here. It is a modeling work that extends the geostrophic balance. Although it does not directly involve SWOT data, it investigates the theory behind the common approximations, including cyclogeostrophc balance. For example, it elucidates the connection between Zhang and Callies [2025] and the cyclogeostrophc balance.**

Dù et al. [2025] might indeed be of interest for some readers, and it refer to SWOT even if it is not at all the core of the paper. We added the reference in the revised manuscript.

**2**

**Line 45: the term "ageostrophic" and "not capture by the balance" is confusing.**

We remove from the revised manuscript the "not capture by the balance" part which is indeed a bit confusing.

**3**

**Line 137: This sentence is confusing. Do you mean it can be written as an L2 norm, without the integral? I do not see how it is used in later "further discussions".**

In the revised manuscript we clarified the sentence and make it explicit in which section of the manuscript the distinction is helpful.

**4**

**Line 236: Here, a discussion of the error of SWOT SSH is warranted. For the purpose of showcasing the method, it is ok to not add noise into the simulated data. But SWOT observation is definitely not going to be the same as the model data. Error or SWOT is discussed in many papers, including Wang et al. [2019], Peral et al. [2024], Zhang and Callies [2025] for a non-exhaustive list.**

We agree that a note on the SWOT data errors is useful. We added one in the revised manuscript, along with references explicitely discussing SWOT errors.

**5**

**(14): for the model case, the sum should also include over the domain points. So N should be much larger than 31.**

As already stated in the text, in Eq. 14 the sum is applied point-wise (spatially) along the time dimension. Therefore, N=31, the number of days in the month of August.

**6**

**Line 342: In fact Tchonang et al. [2025]'s results show cyclogeostrophy did not improve over geostrophic balance. One should be careful when citing.**

Indeed, it was introduced as at some point during the revision process when discussing the regional differences in performance when applying the cyclogesotrophic balance, and we forgot to remove it when using the reference in the eNATL60 section. It has been removed from the revised manuscript.

**7**

**Line 355: Standard error made a sneaky comeback. This is perhaps a typo.**

Indeed, it is fixed in the revised manuscript, along with another typo.

**8**

**Figure 4: Is it correct to call these RMSE? What mean?**

The mean is performed along the time dimension (as explained when introducing Eq. 14). In the revised manuscript we reorganised the sentences of Fig. 4 caption to make it more explicit.